# QUERY EMBEDDING ON HYPER-RELATIONAL KNOWLEDGE GRAPHS

**Dimitrios Alivanistos[1, 4], Max Berrendorf[2], Michael Cochez[1,4], and Mikhail Galkin[3]**
[1] Vrije Universiteit Amsterdam, [2] LMU Munich, [3] Mila, McGill University,
[4] Discovery Lab, Elsevier
`d.alivanistos@vu.nl`, `berrendorf@dbs.ifi.lmu.de`,
`m.cochez@vu.nl`, `mikhail.galkin@mila.quebec`

## ABSTRACT

Multi-hop logical reasoning is an established problem in the field of representation learning on knowledge graphs (KGs). It subsumes both one-hop link prediction as well as other more complex types of logical queries. However, existing algorithms operate only on classical, triple-based graphs, whereas modern KGs often employ a *hyper-relational* modeling paradigm. In this paradigm, typed edges may have several key-value pairs known as *qualifiers* that provide fine-grained context for facts. In queries, this context modifies the meaning of relations, and usually reduces the answer set. Hyper-relational queries are often observed in real-world KG applications, and existing approaches for approximate query answering (QA) cannot make use of qualifier pairs. In this work, we bridge this gap and extend the multi-hop reasoning problem to hyper-relational KGs allowing to tackle this new type of complex queries. Building upon recent advancements in Graph Neural Networks and query embedding techniques, we study how to embed and answer hyper-relational conjunctive queries. Besides that, we propose a method to answer such queries and demonstrate in our experiments that qualifiers improve QA on a diverse set of query patterns.

## 1 INTRODUCTION

Query embedding (QE) on knowledge graphs (KGs) aims to answer logical queries using neural reasoners instead of traditional databases and query languages. Traditionally, a KG is initially loaded into a database that understands a particular query language, e.g., SPARQL. The logic of a query is encoded into *conjunctive graph patterns*, *variables*, and common operators such as *joins* or *unions*.

On the other hand, QE bypasses the need for a database or query engine and performs reasoning directly in a latent space by computing a similarity score between the *query representation* and *entity representations*[1]. A query representation is obtained by processing its equivalent logical formula where joins become intersections ($\wedge$), and variables are existentially quantified ($\exists$). A flurry of recent QE approaches (Hamilton et al., 2018; Ren et al., 2020; Ren & Leskovec, 2020; Kotnis et al., 2020; Arakelyan et al., 2021) expand the range of supported logical operators and graph patterns.

However, all existing QE models work only on classical, triple-based KGs. In contrast, an increasing amount of publicly available (Vrandečić & Krötzsch, 2014; Suchanek et al., 2007) and industrial KGs adopt a *hyper-relational* modeling paradigm where typed edges may have additional attributes, in the form of key-value pairs, known as *qualifiers*. Several standardization efforts embody this paradigm, i.e., RDF* (Hartig, 2017) and Labeled Property Graphs (LPG)[2], with their query languages SPARQL* and GQL, respectively. Such hyper-relational queries involving qualifiers are instances of *higher-order* logical queries, and so far, there has been no attempt to bring neural reasoners to this domain.

In this work, we bridge this gap and propose a neural framework to extend the QE problem to hyper-relational KGs enabling answering more complex queries. Specifically, we focus on logical

---

[1]Existing QE approaches operate only on the entity level and cannot have relations as variables
[2]`https://www.iso.org/standard/76120.html`

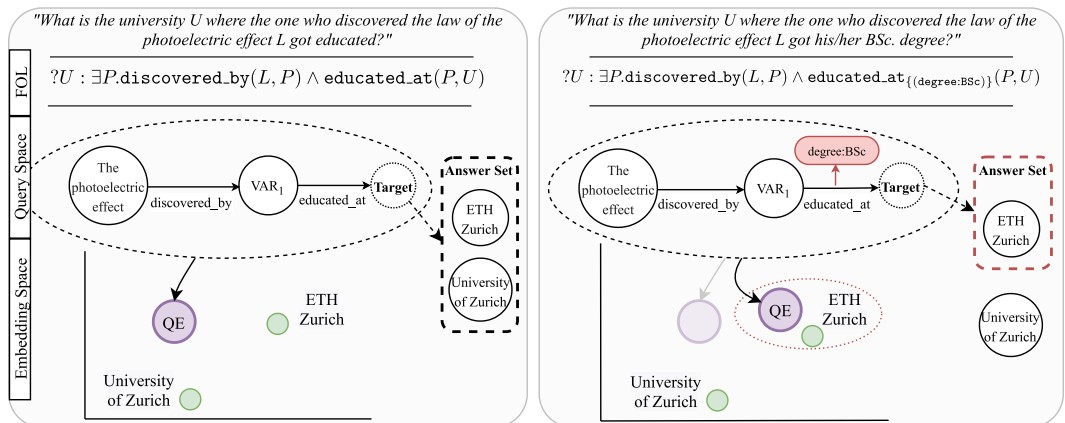

Figure 1: Triple-based (left) and hyper-relational (right) queries. The answer set of the hyper-relational query is reduced with the addition of a qualifier pair, and the final query representation moves closer to the narrowed down answer.

queries that use conjunctions ($\land$) and existential quantifiers ($\exists$), where the function symbols are parameterized with the qualifiers of the relation. This parameterization enables us (cf. Fig. 1) to answer queries like *What is the university U where the one who discovered the law of the photoelectric effect L got his/her BSc. degree?*, which can be written as $?U : \exists P.\texttt{discovered\_by}(L, P) \land \texttt{educated\_at}_{\{(\texttt{degree:BSc})\}}(P, U)$.

Our contributions towards this problem are four-fold. First, as higher-order queries are intractable at practical scale, we show how to express such queries in terms of a subset of first-order logic (FOL) well-explored in the literature. Second, we build upon recent advancements in Graph Neural Networks (GNNs) and propose a method to answer conjunctive hyper-relational queries in latent space. Then, we validate our approach by demonstrating empirically that qualifiers significantly improve query answering (QA) accuracy on a diverse set of query patterns. Finally, we show the robustness of our hyper-relational QA model to a *reification* mechanism commonly used in graph databases to store such graphs on a physical level.

## 2 RELATED WORK

**Query Embedding.** The foundations of neural query answering and query embeddings laid in GQE (Hamilton et al., 2018) considered conjunctive ($\land$) queries with existential ($\exists$) quantifiers modeled as geometric operators based on Deep Sets (Zaheer et al., 2017). Its further extension, QUERY2BOX (Ren et al., 2020), proposed to represent queries as hyper-rectangles instead of points in a latent space. Geometrical operations on those rectangles allowed to answer queries with disjunction ($\lor$) re-written in the Disjunctive Normal Form (DNF). Changing the query representation form to beta distributions enabled BETAE (Ren & Leskovec, 2020) to tackle queries with negation ($\neg$). Another improvement over QUERY2BOX as to answering entailment queries was suggested in EMQL (Sun et al., 2020) by using *count-min sketches.*

The other family of approaches represents queries as Directed Acyclic Graphs (DAGs). MPQE (Daza & Cochez, 2020) assumes variables and targets as nodes in a query graph and applies an R-GCN (Schlichtkrull et al., 2018) encoder over it. It was shown that message passing demonstrates promising generalization capabilities (i.e., when training only on 1-hop queries and evaluating on more complex patterns). Additional gains are brought when the number of R-GCN layers is equal to the graph diameter. Treating the query DAG as a fully-connected (clique) graph, BIQE (Kotnis et al., 2020) applied a Transformer encoder (Vaswani et al., 2017) and allowed to answer queries with multiple targets at different positions in a graph.

Finally, CQD (Arakelyan et al., 2021) showed that it is possible to answer complex queries without an explicit query representation. Instead, CQD decomposes a query in a sequence of reasoning steps and performs a beam search in a latent space of KG embeddings models pre-trained on a simple 1-

hop link prediction task. A particular novelty of this approach is that no end-to-end training on complex queries is required, and any trained embedding model of the existing abundance (Ali et al., 2020; Ji et al., 2020) can be employed as is.

Still, all of the described approaches are limited to triple-based KGs while we extend the QE problem to the domain of hyper-relational KGs. As our approach is based on query graphs, we investigate some of MPQE observations as to query diameter and generalization (see also Appendix G).

**Hyper-relational KG Embedding.** Due to its novelty, embedding hyper-relational KG is a field of ongoing research. Most of the few existing models are end-to-end *decoder-only* CNNs (Rosso et al., 2020; Guan et al., 2020) limited to 1-hop link prediction, i.e., embeddings of entities and relations are stacked and passed through a CNN to score a statement. On the *encoder* side, we are only aware of STARE (Galkin et al., 2020) to work in the hyper-relational setting. STARE extends the message passing framework of CompGCN (Vashishth et al., 2020) by composing qualifiers and aggregating their representations with the primary relation of a statement.

Inspired by the analysis of (Rosso et al., 2020), we take a closer look at comparing hyper-relational queries against their *reified* counterparts (transformed into a triple-only form). We also adopt STARE (Galkin et al., 2020) as a basic query graph encoder and further extend it with attentional aggregators akin to GAT (Veličković et al., 2018).

## 3 Hyper-relational Knowledge Graphs and Queries

**Definition 3.1** (Hyper-relational Knowledge Graph). *Given a finite set of entities $\mathcal{E}$, and a finite set of relations $\mathcal{R}$, let $\mathfrak{Q} = 2^{(\mathcal{R} \times \mathcal{E})}$. Then, we define a hyper-relational knowledge graph as $\mathcal{G} = (\mathcal{E}, \mathcal{R}, \mathcal{S})$, where $\mathcal{S} \subset (\mathcal{E} \times \mathcal{R} \times \mathcal{E} \times \mathfrak{Q})$ is a set of (qualified) statements.*

For a single statement $s = (h, r, t, qp)$, we call $h, t \in \mathcal{E}$ the *head* and *tail* entity, and $r \in \mathcal{R}$ the *(main) relation*. This also indicates the direction of the relation from head to tail. The triple $(h, r, t)$ is also called the *main triple*. $qp = \{q_1, \ldots\} = \{(qr_1, qe_1), \ldots\} \subset \mathcal{R} \times \mathcal{E}$ is the set of *qualifier pairs*, where $\{qr_1, qr_2, \ldots\}$ are the *qualifier relations* and $\{qe_1, qe_2, \ldots\}$ the *qualifier values*.

Hyper-relational KGs extend traditional KGs by enabling to qualify triples. In this work, we solely use hyper-relational KGs and hence use "KG" to denote this variant. The qualifier pair set provides additional information for the semantic interpretation of the main triple $(h, r, t)$. For instance, consider the statement $(\texttt{AlbertEinstein}, \texttt{educated\_at}, \texttt{ETHZurich}, \{(\texttt{degree}, \texttt{BSc})\})$. Here, the qualifier pair $(\texttt{degree}, \texttt{BSc})$ gives additional context on the base triple $(\texttt{AlbertEinstein}, \texttt{educated\_at}, \texttt{ETHZurich})$ (also illustrated on Fig. 1).

Note that this statement can equivalently be written in first order logic (FOL). Specifically, we can write it as a statement with a parameterized predicate $\texttt{educated\_at}_{\{(\texttt{degree:BSc})\}}(\texttt{AlbertEinstein}, \texttt{ETHZurich})$. Then, we note that $\mathfrak{Q}$ is a finite set, meaning that also the number of different parameterizations for the predicate is finite, which means that this becomes a first order logic statement. In this formalism, the KG is the conjunction of all FOL statements. Possible monotonicity concerns are discussed in Appendix E.

We define a hyper-relational query on a hyper-relational KG as follows:[3]

**Definition 3.2** (Hyper-relational Query). *Let $\mathcal{V}$ be a set of variable symbols, and TAR $\in \mathcal{V}$ a special variable denoting the target of the query. Let $\mathcal{E}^+ = \mathcal{E} \uplus \mathcal{V}$. Then, any subset $Q$ of $(\mathcal{E}^+ \times \mathcal{R} \times \mathcal{E}^+ \times \mathfrak{Q})$ is a valid query if its induced graph 1) is a directed acyclic graph, 2) has a topological ordering in which all entities (in this context referred to as anchors) occur before all variables, and 3) TAR must be last in the topological orderings.*[4]

The **answers** $A_{\mathcal{G}}(Q)$ to the query $Q$ are the entities $e \in \mathcal{E}$ that can be assigned to TAR, for which there exist a variable assignment for all other variables occurring in the query graph, such that the instantiated query graph is a subgraph of the complete graph $\mathcal{G}$.

---

[3]The queries considered here are a subset of SPARQL* basic graph pattern queries.

[4]These requirements are common in the literature, and usually stated with formal logic, but not a strict requirement for our approach. See Appendix F for more information.

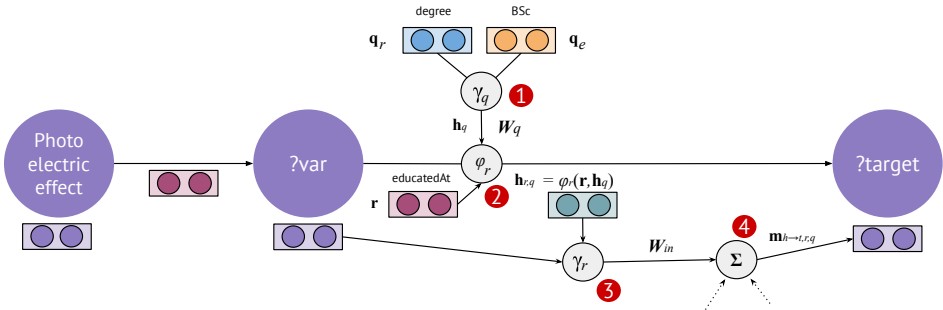

Figure 2: StarE layer intuition: (1) aggregation $\gamma_q$ of qualifiers into a single vector; (2) aggregation $\phi_r$ of a main relation with a qualifiers vector; (3) composition of an enriched relation with the head entity; (4) final message to the tail node.

**Problem Definition.** Given the incomplete KG $\mathcal{G}$ (part of the not observable complete KG $\hat{\mathcal{G}}$) and a query Q. Rank all entities in $\mathcal{G}$ such that answers to the query, if it were asked in the context of the complete KG $\hat{\mathcal{G}}$, are at the top of the ranking.

Since the given KG is not complete, we cannot solve this problem directly as a graph matching problem as usually done in databases. Instead, we compute a latent representation of the query, such that it is close to the embeddings of entities which are the correct answers to the query.

## 4 MODEL DESCRIPTION

Our model is not trained directly on the KG, but rather with a set of queries sampled from it (see Section 5.1). Hence, to describe our model, we consider a query $Q \subset (\mathcal{E}^+ \times \mathcal{R} \times \mathcal{E}^+ \times \mathfrak{Q})$, and describe how we learn to represent it. Our model can be seen as a combination of MPQE (Daza & Cochez, 2020) and StarE (Galkin et al., 2020) with a CompGCN (Vashishth et al., 2020) architecture.

Our model learns representations for entities and the special symbols ({VAR, TAR}), $\hat{\mathbf{E}} \in \mathbb{R}^{(|\mathcal{E}|+2)\times d}$, and relation representations $\mathbf{R} \in \mathbb{R}^{2|\mathcal{R}|\times d}$, where $d$ is a hyper-parameter indicating the embedding dimension. There are two representations for each relation. For a statement $s = (h, r, t, qp) \in Q$, one is used for computing "forward" messages from $h \rightarrow t$, and the other one to compute "backward" messages from $h \leftarrow t$, i.e., inverse relation $r^{-1}$.

We encode the query graph using a sequence of STARE layers (Galkin et al., 2020). First, we select from $\hat{\mathbf{E}}$ the necessary embeddings. These correspond to the ones for all entities in the query, VAR if the query has internal variables, and TAR for targets. The embeddings are then put in $\mathbf{E}$, which also contains one copy of VAR for each unique variable encountered in $Q$.

A layer enriches the entity and relation representations $\mathbf{E}, \mathbf{R}$ to $\mathbf{E}', \mathbf{R}'$ as follows: For each query statement $s = (h, r, t, qp) \in Q$, we compute messages $\mathbf{m_{h \rightarrow t, r, qp}}$ and $\mathbf{m_{h \leftarrow t, r^{-1}, qp}}$. For brevity, we only describe $\mathbf{m_{h \rightarrow t, r, qp}}$, the other direction works analogously.

**Qualifier Representation.** We begin by composing the representations of the components of each qualifier pair $q_i = (qr_i, qe_i) \in qp$ into a single representation: $\mathbf{h_{q_i}} = \gamma_q(\mathbf{E}[qe_i], \mathbf{R}[qr_i])$, where $\gamma_q$ denotes a composition function (Vashishth et al., 2020), e.g., the Hadamard product, and we use $\mathbf{X}[y]$ to indicate the representation of entity y projected in vector space X e.g $qe_i$ in $E$. Next, we aggregate all qualifier pair representations for $qp$, as well as the representation of the main relation $r$ into a qualified relation representation using an aggregation function $\phi_r$: $\mathbf{h_{r, qp}} = \phi_r(\mathbf{R}[r], \{\mathbf{h_{q_i}}\}_{q_i \in qp})$. For $\phi_r$ we experiment with a simple sum aggregation, and an attention mechanism.

**The Message-Passing Step.** To obtain a message $\mathbf{m_{h \rightarrow t, r, qp}}$, the qualified relation representation is further composed with the source entity representation $\mathbf{E}[h]$ using another composition function $\gamma_r$. The result gets linearly transformed by $\mathbf{W_{\rightarrow}}$, a layer specific trainable parameter: $\mathbf{m_{h \rightarrow t, r, qp}} = \mathbf{W_{\rightarrow}} \gamma_r(\mathbf{E}[h], \mathbf{h_{r, qp}})$. For inverse relations, we use a separate weight $\mathbf{W_{\leftarrow}}$. Next, we aggregate all incoming messages $\mathbf{a_{t, \rightarrow}} = \phi_m(\{\mathbf{m_{h \rightarrow t', r, qp}} \mid (h, r, t', qp) \in Q, t = t'\})$ us-

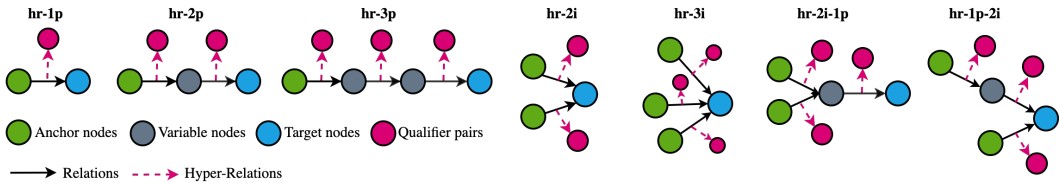

Figure 3: The hyper-relational formulas and their graphical structures. The qualifier pairs attached to each edge may vary in $0..n$; for brevity we represent them as a single pair. We also allow qualifiers to exist only on *certain* edges of a query, i.e., not all edges necessarily contain them.

ing a message aggregation function $\phi_m$, e.g., a (weighted) sum. Besides computing these message aggregates in each direction, we also compute a self-loop update $\mathbf{a}_{\mathbf{e},\circlearrowleft} = \mathbf{W}_{\circlearrowleft}\gamma_r(\mathbf{e}, \mathbf{r}_{\circlearrowleft})$, where $\mathbf{W}_{\circlearrowleft}, \mathbf{r}_{\circlearrowleft}$ are trainable parameters. The updated entity representation is then obtained as the average over both directions and the self loop, with an additional activation $\sigma$ applied to it: $\mathbf{E}'[e] = \sigma\left(\frac{1}{3}(\mathbf{a}_{\mathbf{e},\circlearrowleft} + \mathbf{a}_{\mathbf{e},\rightarrow} + \mathbf{a}_{\mathbf{e},\leftarrow})\right)$ Finally, the relation representations are updated by a linear transformation $\mathbf{R}'[r] = \mathbf{W}_{\mathbf{r}}\mathbf{R}[r]$, where $\mathbf{W}_{\mathbf{r}}$ is a layer specific trainable weight.

**Query Representation.** After applying multiple STARE layers, we obtain enriched node representations $\mathbf{E}^*$ for all nodes in the query graph. As final query representation $\mathbf{x_Q}$, we aggregate all node representations of the query graph, $\mathbf{x_Q} = \phi_q(\{\mathbf{E}^*[e] \mid e \in \mathcal{E}^+ \wedge ((e, r, t, qp) \in Q \vee (h, r, e, qp) \in Q)\})$, with $\phi_q$ denoting an aggregation function, e.g., the sum. Alternatively, we only select the final representation of the unique target node, $\mathbf{x_Q} = \mathbf{E}^*[\text{TAR}]$. To score answer entity candidates, we use the similarity of the query representation and the entity representation, i.e., $score(Q, e) = sim(\mathbf{x_Q}, \mathbf{E}^*[e])$, such as the dot product, or cosine similarity.

We designate the described model as STARQE (*Query Embedding for RDF Star Graphs*) since RDF* is one of the most widely adopted standards for hyper-relational KGs.

## 5 EXPERIMENTS

In this section, we empirically evaluate the performance of QA over hyper-relational KGs. We design experiments to tackle the following research questions: **RQ1)** Does QA performance benefit from the use of qualifiers? **RQ2)** What are the generalization capabilities of our hyper-relational QA approach? **RQ3)** Does QA performance depend on the physical representation of a hyper-relational KG, i.e., *reification*?

### 5.1 DATASET

Existing QE datasets based on Freebase (Toutanova & Chen, 2015) and NELL (Carlson et al., 2010) are not applicable in our case since their underlying KGs are strictly triple-based. Thus, we design a new hyper-relational QE dataset based on WD50K (Galkin et al., 2020)[5] comprised of Wikidata statements, with varying numbers of qualifiers.

**WD50K-QE.** We introduce hyper-relational variants of 7 query patterns commonly used in the related work (Hamilton et al., 2018; Ren et al., 2020; Sun et al., 2020; Arakelyan et al., 2021) where each edge can have $[0, n]$ qualifier pairs. The patterns contain *projection* queries (designated with -p), *intersection* queries (designated with -i), and their combinations (cf. Fig. 3). Note that the simplest 1p pattern corresponds to a well-studied link prediction task. Qualifiers allow more flexibility in query construction, i.e., we further modify formulas by conditioning the existence of qualifiers and their amount over a particular edge. We include dataset statistics and further details as to dataset construction in Appendices B and D.

These patterns are then translated to the SPARQL* format (Hartig, 2017) and used to retrieve materialized query graphs from specific graph splits. Following existing work, we make sure that validation and test queries contain at least one edge unseen in the training queries, such that evaluated models have to predict new links in addition to QA. As we are in the transductive setup where

---

[5]This dataset is available under CC BY 4.0.

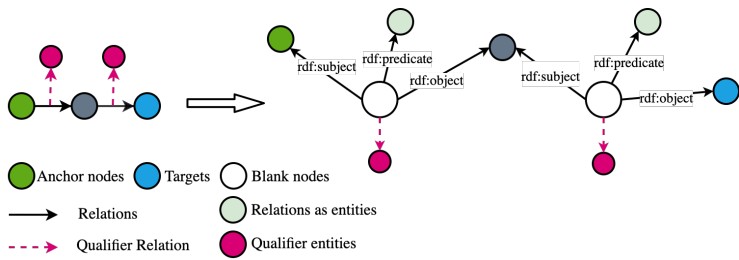

Figure 4: Example of a reification process: An original `hr-2p` query pattern (left) is reified through *standard RDF reification* (right). Note the change of the graph topology: the reified variant has two blank nodes, three new pre-defined relation types, and original edge types became nodes connected via the `rdf:predicate` edge. The distance between the anchor and target increased, too.

all entities and relation types have to be seen in training, we also ensure this for all entity and relation types appearing in qualifiers.

Due to the (current) lack of standardized data storage format for hyper-relational (RDF*) graphs in graph databases, particular implementations of RDF* employ *reification*, i.e, transformation of hyper-relational statements as defined in Section 3 to plain triples. Reification approaches (Frey et al., 2019) introduce auxiliary virtual nodes, new relation types, turn relations into nodes, and might dramatically change the original graph topology. An example of a *standard RDF reification* (Brickley et al., 2014) in Fig. 4 transforms an original `hr-2p` query with three nodes, two edges, and two qualifier pairs into a new graph with nine nodes and eight edges with rigidly defined edge types. However, the logical interpretation of a query remains the same. Therefore, we want hyper-relational QE models to be robust to the underlying graph topology. For this reason, we ship dataset queries in both formats, i.e., hyper-relational RDF* and reified with triples, and study the performance difference in a dedicated experiment.

## 5.2 EVALUATION PROTOCOL

In all experiments, we evaluate the model in the rank-based evaluation setting. Each query is encoded into a query embedding vector $\mathbf{x_q} \in \mathbb{R}$. We compute a similarity score $sim(\mathbf{x_q}, \mathbf{E}[e])$ between the query embedding and the entity representation for each entity $e \in \mathcal{E}$. The rank $r \in \mathbb{N}^+$ of an entity is its position in the list of entities sorted decreasingly by score. We compute *filtered* ranks (Bordes et al., 2013), i.e., while computing the rank of a correct entity, we ignore the scores of other correct entities. We resolve exactly equal scores using the *realistic rank* (Berrendorf et al., 2020), i.e., all entities with equal score obtain the rank of the average of the first and last position.

Given a set of individual ranks $\{r_i\}_{i=1}^{n}$, we aggregate them into a single-figure measure using several different aggregation measures: The Hits@k (H@k) metric measures the frequency of ranks at most $k \in \mathbb{N}^+$, i.e., $H@k = \frac{1}{n}\sum \mathbb{I}[r_i \leq k]$, with $\mathbb{I}$ denoting the indicator function. H@k lies between zero and one, with larger values showing better performance. The Mean Reciprocal Rank (MRR) is the (arithmetic) mean over the reciprocal ranks, i.e., $MRR = \frac{1}{n}\sum r_i^{-1}$, with a value range of $(0, 1]$, and larger values indicating better results. It can be equivalently interpreted as the inverse harmonic mean over all ranks, and thus is stronger influenced by smaller ranks.

Since the size of the answer set of queries varies greatly,[6] queries with large answer sets would strongly influence the overall score compared to very specific queries with a single entity as answer. In contrast to rank-based evaluation in existing QE work, we propose to weight each rank in the aforementioned averages by the inverse cardinality of the answer set to compensate for their imbalance. Thereby, each query contributes an equal proportion to the final score.

## 5.3 HYPER-RELATIONAL QA

As we are the first to introduce the problem of hyper-relational QA, there is no established baseline available at the time of writing. Hence, we compare our method to several alternative approaches.

---

[6]For, e.g., `hr-2p` we observe a maximum answer set cardinality of 1,351, while the upper quartile is 3.

Table 1: QA performance of STARQE and the baselines when training on all hyper-relational query patterns. We omit the `hr-` prefix for brevity. Best results (excluding the Oracle) are marked in **bold**.

| Pattern | 1p | 2p | 3p | 2i | 3i | 2i-1p | 1p-2i |
|---|---|---|---|---|---|---|---|
| | Hits@10 (%) | | | | | | |
| StarQE | 51.72 | **51.20** | **65.50** | 77.78 | 92.64 | **61.81** | **81.60** |
| Triple-Only | 45.04 | 12.76 | 24.66 | 69.74 | 91.74 | 16.77 | 40.67 |
| Reification | **55.17** | 50.86 | 63.65 | **81.25** | **95.31** | 61.05 | 80.49 |
| Zero Layers | 44.93 | 29.94 | 38.45 | 67.79 | 90.66 | 35.48 | 47.85 |
| Oracle | 81.03 | 24.11 | 38.47 | 95.54 | 99.67 | 32.74 | 76.96 |
| | MRR (%) | | | | | | |
| StarQE | 30.98 | **44.13** | **52.96** | **63.14** | 83.78 | **55.20** | **71.52** |
| Triple-Only | 22.25 | 6.73 | 14.05 | 48.01 | 74.52 | 8.16 | 22.23 |
| Reification | **34.78** | 43.42 | 50.77 | 61.01 | **85.17** | 49.09 | 64.21 |
| Zero Layers | 27.55 | 19.10 | 21.25 | 50.27 | 80.62 | 24.49 | 30.04 |
| Oracle | 79.16 | 18.40 | 23.72 | 90.43 | 97.91 | 21.10 | 54.74 |

**Implementation.** We implement STARQE[7] and other baselines in PyTorch (Paszke et al., 2019) (MIT License). We run a hyperparameter optimization (HPO) pipeline on a validation set for each model and report the best setup in the Appendix J. All experiments are executed on machines with single GTX 1080 Ti or RTX 2080 Ti GPU and 12 or 32 CPUs.

**Triple-Only** For the first experiment, we remove all qualifiers from the hyper-relational statements in the query graph leaving the base triples $(h, r, t)$. Thus, we isolate the effect of qualifiers in answering the queries correctly. Note that in effect, this is similar to the MPQE approach, but with a better GNN and aggregation. To do this, we have to retain the same queries as in other experiments, but remove the qualifiers upon loading. The set of targets for these queries remains unchanged, e.g., in the hyper-relational query from Fig. 1 we would drop the (degree:BSc) qualifier but still have only ETHZurich as a correct answer.

**Reification.** For the second setting, we convert the hyper-relational query graph to plain triples via reification (see Section 5.1). The effects of such a transformation include the addition of two extra nodes per triple in the query, to represent blank nodes and predicates (more details in Appendix A). Here, we investigate whether STARQE is able to produce the same semantic interpretation of a topologically different query. Note that, while conceptually the default relation enrichment mechanism of STARQE resembles *singleton property reification* (Frey et al., 2019), its semantic interpretation is equivalent to *standard RDF reification.*

**Zero Layers.** To measure the effect of message passing, we consider a model akin to *bag-of-words*, which trains entity and relation embeddings without message passing before graph aggregation.

**Oracle.** If we take away the qualifier information, we could compare with several QE approaches (e.g. GQE, MPQE, Query2box, BetaE, EmQL, and CQD). The Oracle setup, is not just an upper bound to these, but to all possible, non-hyper-relational QE models. It simulates the best possible QE model that has perfect link prediction and ranking capabilities, but without the ability to use qualifier information. Using this setting we investigate the impact of qualifier information on our queries. More details are in Appendix I.

**Discussion.** The results shown in Table 1 indicate that STARQE is able to tackle hyper-relational queries of varying complexity. That is, the performance on complex intersection and projection queries is often higher than that of a simple link prediction (`hr-1p`). Particularly, queries with intersections (`-i`) demonstrate outstanding accuracy, but it should be noted here that also the Oracle setting performs very well in this case. Importantly, MRR values are relatively close to Hits@10 which means that more precise measures like Hits@3 and Hits@1 retain good performance (we provide a detailed breakdown in Appendix J).

---

[7]STARQE implementation: https://github.com/DimitrisAlivas/StarQE

Table 2: Results for the generalization experiment. Colored cells denote training query patterns for each style, e.g., in the EMQL-style we train only on `1p` and `2i` patterns and evaluate on all. The `hr-` prefix is omitted for brevity.

| Evaluation Style | 1p | 2p | 3p | 2i | 3i | 2i-1p | 1p-2i |
|---|---|---|---|---|---|---|---|
| | Hits@10 (%) | | | | | | |
| StarQE-like | 51.72 | **51.20** | 65.50 | 77.78 | 92.64 | **61.81** | **81.60** |
| Q2B-like | 55.44 | 51.10 | **66.39** | 78.79 | 94.20 | 57.49 | 80.49 |
| EmQL-like | 50.10 | 16.45 | 44.36 | 75.86 | 93.55 | 6.79 | 62.80 |
| MPQE-like | 48.48 | 12.57 | 34.19 | 83.04 | 96.32 | 14.75 | 61.02 |
| MPQE-like + Reif | **58.43** | 12.02 | 31.14 | **83.77** | **97.22** | 13.50 | 50.92 |
| | MRR (%) | | | | | | |
| StarQE-like | 30.98 | **44.13** | **52.96** | **63.14** | 83.78 | **55.20** | **71.52** |
| Q2B-like | 33.04 | 41.99 | 51.71 | 61.72 | 83.49 | 44.24 | 67.04 |
| EmQL-like | 32.01 | 10.09 | 27.94 | 61.45 | **86.28** | 3.73 | 53.58 |
| MPQE-like | 26.83 | 6.79 | 19.72 | 56.16 | 74.35 | 9.62 | 39.81 |
| MPQE-like + Reif | **36.36** | 6.12 | 17.11 | 56.81 | 77.29 | 8.32 | 29.25 |

To investigate if this performance could be attributed to the impact of qualifiers, we run a *Triple-Only* and Oracle setup. These experiments show that for some query patterns, qualifiers play an important role. Without them, we would not be able to get good results for the `2p`, `3p`, `2i-1p` and `1p-2i` queries. The reason that the Oracle cannot answer these well is that despite its access to the test set, the mistakes it makes accumulate when there are more hops akin to beam search (recall that the Oracle does not have access to qualifiers and considers all edges with a given relation) . For queries that only involve one reasoning step, we notice that the Oracle can, and hence a normal link predictor might be able, to perform very well. When more paths intersect (`2i` and `3i`), the chance of making a mistake goes down. This observation is similar what can be seen in e.g., CQD (Arakelyan et al., 2021), and can be ascribed to each of the different paths constraining the possible answer set, while cancelling out mistakes. More experiments on qualifiers impact are reported in Appendix H.

We observe a comparative performance running the *Reification* baseline. It suggests that our QE framework is robust to the underlying graph topology retaining the same logical interpretation of a complex query. We believe it is a promising sign of enabling hyper-relational QA on a wide range of physical graph implementations.

Finally, we find that message passing layers are essential for maintaining high accuracy as the *Zero Layers* baseline lags far behind GNN-enabled models. One explanation for this observation can be that without message passing, variable nodes do not receive any updates and are thus not "resolved" properly. To some extent counter-intuitively, we also observed that it does not make a large difference whether relation embeddings are included in the aggregation or not. Relatively high performance on `1p`, `2i`, `3i` queries can be explained by their very specific *star-shaped* query pattern which is essentially 1-hop with multiple branches joining at the center node.

## 5.4 GENERALIZATION

Following the related work, we experiment with how well our approach can generalize to complex query patterns if trained on simple ones. Note that below we understand all query patterns as hyper-relational, i.e., having the `hr-` prefix. There exist several styles for measuring generalization in the literature that we include in the experiment:

**Q2B-like.** The style is used by QUERY2BOX (Ren et al., 2020) and assumes training only on `1p,2p,3p,2i,3i` queries while evaluating on two additional patterns `2i-1p`, `1p-2i`.

**EmQL-like.** The other approach proposed by EMQL (Sun et al., 2020) employs only `1p`, `2i` patterns for training, using five more complex ones for evaluation.

**MPQE-like.** The hardest generalization setup used in MPQE (Daza & Cochez, 2020) allows training only on `1p` queries, i.e., vanilla link prediction, while all evaluation queries include unseen intersections and projections.

**MPQE-like + Reif.** To measure the impact of reification on generalization, we also run the experiment on reified versions of all query patterns. Similarly to *MPQE-like*, this setup allows training only on `1p` reified pattern and evaluates the performance on other, more complex reified patterns.

**Discussion.** Table 2 summarizes the generalization results. As reference, we include the STARQE results in the no-generalization setup when training and evaluating on all query patterns. Generally, we observe that all setups generalise well on intersection queries (`-i`) even when training in the most restricted (`1p`) mode. The *Q2B-like* regime demonstrates appealing generalization capabilities on (`2i-1p, 2p-1i`), indicating that it is not necessary to train on all query types. However, moving to a fine-grained study of most impactful patterns, we find that projection (`-p`) patterns are rather important for generalization, as both *EmQL* and *MPQE* styles dramatically fall behind in accuracy, especially in the MRR metric indicating that higher precision results are impaired the most.

The *MPQE* style is clearly the hardest when having only one training pattern. The higher results on intersection (`-i`) patterns can be explained by a small cardinality of the answer set, i.e., qualifiers make a query very selective with very few possible answers. Finally, it appears that reification (*MPQE+ Reif*) impedes generalization capabilities and overall accuracy according to the MRR results. This can be explained by the graph topologies produced when reifying complex queries, and training only on `1p` is not sufficient.

# 6 LIMITATIONS & FUTURE WORK

In this section, we discuss limitations of the current work, and future research directions. The first limitation of our work is that we do not allow literal values like numbers, text, and time in our graph. This means that we cannot, for example, handle queries asking for people born in the year 1980. These and more complex values can be incorporated as node features (Wilcke et al., 2020).

Secondly, more logical operators, such as negation (which could be included as a qualifier), disjunctions, cardinality constraints, etc. can be considered. In this work, we only allow variables in the head and tail positions. Nonetheless, one can also formulate queries with variables in more diverse positions, e.g., qualifier value or main relation.

Moreover, the current work allows for different query shapes compared to prior work because queries are not limited to DAGs (see Appendix F for details). However, our work does not allow all shapes. Specifically, it is currently unclear how queries with cycles would behave.

Another future direction can be found in the many operators which query languages like SPARQL have. For example, queries including paths, aggregations, sub-queries, filters on literals, etc. Further research work is required towards explainability. Currently, our system does not provide explanations to the answers. An initial direction is to analyse the intermediate values in variable positions which can be used as explanations, but likely need to be explicitly trained to behave that way. Finally, an interesting research direction is the use of our approach for the creation of query plans.

# 7 CONCLUSION

In this work, we have studied and addressed the extension of the multi-hop logical reasoning problem to hyper-relational KGs. We addressed the theoretical considerations of having qualifiers in the context of QA and discussed the effects it can have on it, such as cardinality of the answer set. We proposed the first hyper-relational QE model, STARQE, based on a GNN encoder to work in this new setup. We introduced a new dataset, WD50K-QE, with hyper-relational variants of 7 well studied query patterns and analysed the performance of our model on each of them. Our results suggest that qualifiers help in obtaining more accurate answers compared to triple-only graphs. We also demonstrate the robustness of our approach to structural changes involved in the process of reification. Finally, we evaluate the generalisation capabilities of our model in all settings and find that it is able to accurately answer unseen patterns.

**Reproducibility Statement.** The experimental setup and implementation details are described in Section 5. We elaborate on the dataset construction process in Appendices B and C. All hyperparameters are listed in Table 6. The complete source code is available for reviewers in the supplementary material, and will be made available open source. More experimental evidence is provided in Appendix H and more detailed metrics per query pattern are listed in Appendix J.

**Ethics Statement.** The assumption of our approach as well as related techniques is that the data used for training solely contains true facts. If this data is (intentionally) biased or erroneous to start with, further biased or wrong information will be derived, which could lead to harmful conclusions and decisions. Besides, even if the input data to the system is correct, the system could due to its imperfections still derive incorrect answers to queries, which might, again lead to wrong conclusions. Hence, users of this technology must be made aware that answers to their queries are always approximate, and this technology ought not to be used in a process where correctness is critical.

**Acknowledgements.** This work has been funded by the Elsevier Discovery Lab and the German Federal Ministry of Education and Research (BMBF) under Grant No. 01IS18036A. The authors of this work take full responsibilities for its content. For performing the experiments the authors made use of the LISA system provided by SurfSara and the DAS cluster (Bal et al., 2016).

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

## APPENDIX

## A    TYPES OF GRAPHS

In this section, we discuss in further detail the different approaches to break free from the restriction of pairwise relations. The intuition how those approaches are different is presented in Fig. 5.

**Hyper-relational Graphs**    KGs used in our work are a subset of hyper-relational graphs. In general, hyper-relational graphs can also contain literal values, like integers and string values as qualifier values. Labeled property graphs are another example of a subset of hyper-relation graphs, but they typically only allow literals as qualifier values. The Wikidata model (Erxleben et al., 2014) and RDF* (Hartig, 2017) allows for both literals and entities as values. Many implementations of RDF* do make the monotonicity assumption, which we also made in the paper. Some will also merge statements in case they have the same main triple. The resulting statement will have the same main triple, but as qualifier information, the union of the sets of qualifier pairs of the original statements.

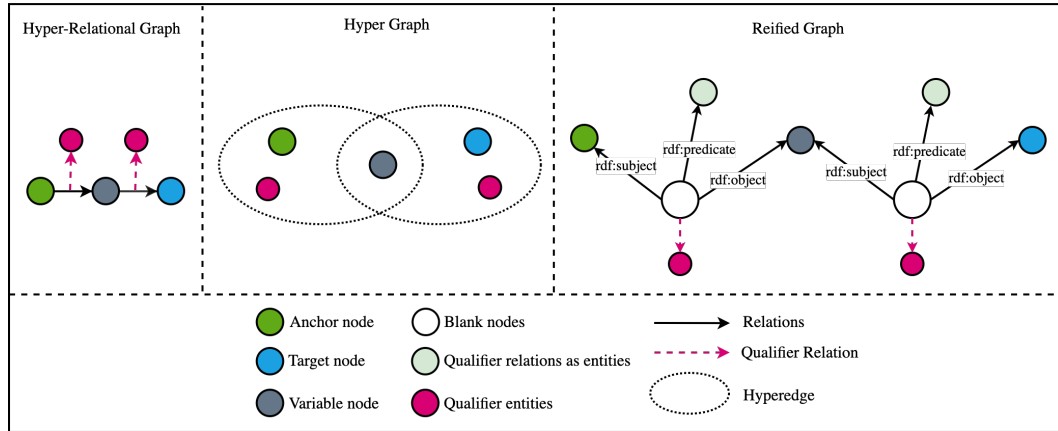

Figure 5: The same query expressed in three different approaches. Hyper-relational (left), hypergraph (middle), and reified (right) graphs.

**Hypergraphs**  Hypergraphs are another type of graphs, initially proposed by Berge (Berge, 1984), where *hyperedges* link one or more vertices. These hyperedges group together sets of nodes in a relation, but lose information of the particular roles that entities play in this relation. Besides, each different composition of elements in the relation introduces a new hyperedge type, causing a combinatorial explosion of these types.

**Reified Graphs**  Hyper-relational and reified graphs have equivalent expressive power compared to labelled directed graphs. The reason is that we can define a bijection between these graphs. However, depending on the use case, the different graph types have benefits.

When we convert a hyper-relational graph to an RDF graph, we end up with what is called the reified graph. There are multiple approaches on how to perform this conversion. In our work, we use Standard RDF Reification[8]. It introduces new, so called *blank* nodes to represent statements with all parts of a statement attached. This allows for all information to exist on the same level. One of the disadvantages to this approach is the addition of auxiliary nodes, which heavily affect the structure of the graph and quickly inflate the graph size.

## B  THE WD50K-QE DATASET

In this section, we describe the generation of the hyper-relational queries that we use for training, validation and testing of our models. We start with WD50K (Galkin et al., 2020), which is a WikiData-based dataset created for hyper-relational link prediction. This dataset already provides with train, validation and test splits, each containing a selection of hyper-relational triples. It is publicly available by the authors, in CSV format. In order to utilise it for our work, we converted it from CSV to RDF*.

The overall pipeline is presented on Fig. 6. To generate the queries, we hosted the converted dataset on a graph database with support for hyper-relational data. For our use-case, we use anzograph[9]. We utilize 3 named graphs[10]: *triple_train*, *triple_validation*, and *triple_test*, to prevent validation and test set leakage. In the evaluation of approximate QA, it is common to have queries with **at least one unseen edge** from the test set, which is also applied upon our query generation process. We ensure that:

- For the training set, all statements in a query come from the *triple_train* set (only).
- For the validation set, one statement comes from the *triple_validation* and the other edge(s) come from either *triple_train* or *triple_validation*, and

---

[8] https://www.w3.org/TR/2014/REC-rdf11-mt-20140225/#reification
[9] https://www.cambridgesemantics.com/anzograph/
[10] https://www.w3.org/TR/rdf11-concepts/#section-dataset

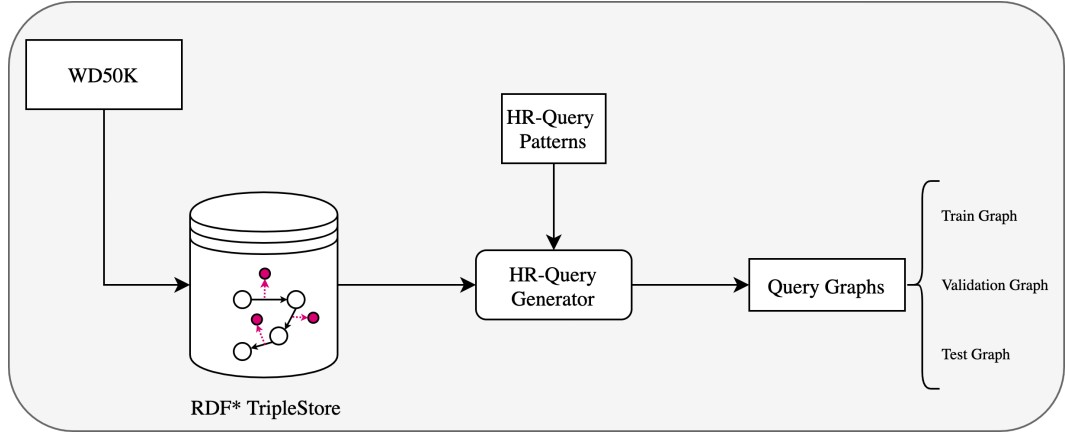

Figure 6: A diagram of the query generation process. On the far left, the chosen dataset (WD50K) is uploaded in the RDF*-compatible triplestore (AnzoGraph). As a follow up, we translated the hyper-relational query patterns to SPARQL* and executed them against the triplestore to retrieve the desired splits.

| Pattern | 1p | 2p | 3p | 2i | 3i | 2i-1p | 1p-2i |
|---|---|---|---|---|---|---|---|
| train | 24,819 | 313,088 | 5,950,990 | 48,513 | 318,735 | 306,022 | 1,088,539 |
| validation | 4,100 | 100,706 | 2,968,315 | 15,648 | 169,195 | 169,438 | 569,957 |
| test | 7,716 | 202,045 | 6,433,476 | 38,207 | 547,272 | 445,007 | 1,267,452 |

Table 3: The amount of queries for each of the different query patterns. We notice that for some patterns there are many more queries than for others, which is also why we report results for the patterns separately.

- For the test set, one statement comes from *triple_test*, and the other statement from any of *triple_train*, *triple_validation*, and *triple_test*

In order to generate the query data splits, we constructed SPARQL* queries that correspond to a specific pattern. We call these higher order queries *formulas*. Using these formulas to generate queries has several advantages compared to other approaches found in related work. In the related work, we encounter sampling using techniques such as random walks to create queries which have the shape according to the pattern. With our approach, we have more control over what the samples are, and thus a finer control of the inputs to our experiments.

Moreover, we can also ensure that we do not sample with replacement, we are not biased towards specific high degree nodes, and at the same time guarantee that we do not sample queries that are isomorphic with each other. A final benefit is that with our approach, we can immediately retrieve all answers for a query, instead of only one.

One final issue we encountered had to do with the nodes that have a high in-degree in our dataset. Their existence causes a skewing of the distribution of correct answers to queries, and has to be resolved. In Appendix C, we describe this issue and explain our approach to overcome it.

As a result of this procedure, we obtain **all** queries of the given shapes, and are guaranteed that there are no duplicates nor isomorphism among the queries. The queries used in this paper are such that every edge in the query has one qualifier, which means that, in the knowledge graph, the same edge has at least one, but possibly more qualifiers. The amount of queries for the different patterns can be found in Table 3.

## C Issues Sampling Queries with Joins in a Graph with High Degree Nodes

In our query generation step, we exclude high in-degree nodes as a target for specific queries. In this section, we explain why that choice was made. We will focus on the `3i` pattern. The same argument holds for other patterns with joins, like `2i`, `2i-1p`, `1p-2i`.

The queries are randomly sampled from all possible queries that can be formed by matching the pattern with the data graph. For the `3i` pattern, a node with an in-degree of $n$, results in $\binom{n}{3}$ different queries, with that node as a target.

The problem is that if we randomly sample our queries, the answer would usually be one of the highest degree nodes. Concretely, in our graph data, the highest observed in-degree is 4,424, which would result in 14,421,138,424 possible queries with the corresponding node as an answer. If we generated all possible queries, we would end up with 38,011,148,464 different ones. So, the node with the highest in-degree is already responsible for 38% of the queries, meaning that system answering 3-i queries that are randomly sampled, would get 38% correct by always giving that answer. Moreover, if we look at the Hits@10 metric, we would score 93%, by always predicting the same ranking (the top 10 in descending frequency). Note that existing query embedding models have been evaluated like this in the past, ignoring this data issue.

Hence, we decided to make the task harder by removing these high degree nodes for queries with joins. That is, by putting the threshold at an in-degree of 50, we remove the 623 nodes with the highest in-degree which would have represented 99.9% of the possible answers for the `3i` queries without this modification. After filtering, the baseline of always predicting the same ranking for the randomly sampled queries, will result in a Hits@10 of 3% in the best case.

For other query shapes where a join is involved, the situation is similar, but less pronounced.

## D Size of the Answer Set of a Query with more Qualifiers

When we have a query with or without qualifiers, and we add more qualifiers, the number of answers to this query can only become smaller. Intuitively, this happens because the query becomes more specific. In this section we prove this intuition correct. Note that proving this requires monotonicity, as we defined in the paper. Without this assumption - for example allowing non-monotonic qualifier relations - would result in losing the guarantee that the answer set becomes smaller.

Given a query $Q$, and a query $\overline{Q}$ that is the same, except for one set of qualifier pairs of $\overline{Q}$ which can have extra pairs, then the answers to $\overline{Q}$ are a subset of the answers to $Q$.

Formally:

**Theorem D.1.** *Given a KG $\mathcal{G} = (\mathcal{E}, \mathcal{R}, \mathcal{S})$, where $\mathcal{S} \subset (\mathcal{E} \times \mathcal{R} \times \mathcal{E} \times \mathfrak{Q})$, a query $Q = \{(h_1, r_1, t_1, qp_1), \dots (h_n, r_n, t_n, qp_n)\}$, and a second query $\overline{Q} = \{(h_1, r_1, t_1, \overline{qp_1}), \dots (h_n, r_n, t_n, \overline{qp_n})\}$, where $\exists! k : qp_k \subseteq \overline{qp_k}$ and $\forall x((x \in [1, \dots n] \wedge x \neq k) \rightarrow \overline{qp_x} = qp_x)$. Then, $A_{\mathcal{G}}(\overline{Q}) \subseteq A_{\mathcal{G}}(Q)$.*

*Proof.* Assuming symbols as defined in the theorem, we show that $\forall a : a \in A_{\mathcal{G}}(\overline{Q}) \implies a \in A_{\mathcal{G}}(Q)$. If $a$ is an answer to $\overline{Q}$, then its associated variable substitution $v$ is such that $(v(h_k), r_k, v(t_k), \overline{qp_k}) \in \mathcal{G}$. From monotonicty (see Appendix E) , and given $qp_k \subseteq \overline{qp_k}$ we then know that $(v(h_k), r_k, v(t_k), qp_k) \in \mathcal{G}$. And hence, using the same variable substitution, $a$ is an answer for $Q$. $\qquad\square$

**Corollary D.1.** *By induction, given a query $Q$, and a query $\overline{Q}$ that is the same, except for **any** set of qualifier pairs of $\overline{Q}$ which can have extra pairs, then the answers to $\overline{Q}$ are a subset of the answers to $Q$.*

**Corollary D.2.** *As a special case, given a query $Q$, and a query $\overline{Q}$ that is the same, except that $Q$ does not have qualifiers, while $\overline{Q}$ can have qualifier pairs on its statements, then the answers to $\overline{Q}$ are a subset of the answers to $Q$.*

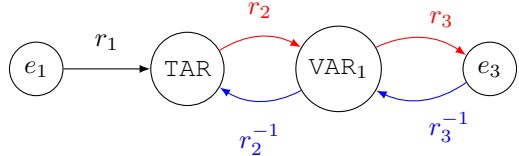

Figure 7: Visualization for Appendix F. Original query (black and red) violating the original definition of a valid query graph, and equivalently transformed query (black and blue) which fulfils the properties becoming `1p-2i` query.

## E   ON MONOTONICITY

In this work, we restrict the qualifiers to respect monotonicity in the KG in the following sense:

$$(h, r, t, qp) \in \mathcal{G} \wedge qp' \subseteq qp \implies (h, r, t, qp') \in \mathcal{G}$$

This implies that if a qualified statement in the KG has a set of qualifier pairs, then the KG also contains the qualified statement with any subset thereof. As a result, some types of qualifying information cannot be used, e.g., a qualifier indicating that a relation does not hold (i.e., negation). This breaks monotonicity since the existence of such a statement would further imply the existence of that statement without the qualifier, leading to contradiction.

## F   RELAXING THE REQUIREMENTS ON QUERIES

In the paper, we limited our queries with the same limitations as done in prior work. Here we will discuss why not all of these restrictions are needed for our work, and how our evaluation already includes some of these more general cases.

As a reminder the definition of our queries is as follows:

**Definition F.1** (Hyper-relational Query). *Let $\mathcal{V}$ be a set of variable symbols, and $TAR \in \mathcal{V}$ a special variable denoting the* target *of the query. Let $\mathcal{E}^+ = \mathcal{E} \uplus \mathcal{V}$. Then, any subset $Q$ of $(\mathcal{E}^+ \times \mathcal{R} \times \mathcal{E}^+ \times \mathfrak{Q})$ is a valid query if its induced graph*

*1)  is a directed acyclic graph,*

*2)  has a topological ordering in which all entities (in this context referred to as anchors) occur before all variables, and*

*3)  $TAR$ must be last in the topological orderings.*

The main reason why we deal with more general queries is because our query encoder learns representation for both normal and inverse relations. We use $r^{-1}$ to indicate the inverse of relation $r$. When encoding a query with a (normal) relation $r$, then both representations $\mathbf{R}[r]$ and $\mathbf{R}[r^{-1}]$ are used simultaneously. If we encounter a query using the inverse relation $r^{-1}$, we can use that inverse relation $\mathbf{R}[r^{-1}]$ and the inverse of the inverse $\mathbf{R}[(r^{-1})^{-1}] = \mathbf{R}[r]$, or in other words the normal relation representation of $r$. This means that even if we only ever train with the normal relation, we have the ability to invert relations, and hence we can lift some limitations.

For example, let us look at the query

$$Q_{\text{new\_pattern}} = \{(e_1, r_1, TAR, \{\}), (TAR, r_2, VAR_1, \{\}), (VAR_1, r_3, e_3, \{\})\},$$

as shown in Fig. 7. This query breaks two of the requirements. It has an entity $e_3$ occurring after variables $VAR_1$ and $TAR$ in the topological ordering. And, $TAR$ is not in the last position.

However, using inverse relations, we can convert this query into:

$$Q_{\text{known\_pattern}} = \left\{(e_1, r_1, TAR, \{\}), (VAR_1, r_2^{-1}, TAR, \{\}), (e_3, r_3^{-1}, VAR_1, \{\})\right\}.$$

This transformation converts the query into a `1p-2i` query, which is a pattern among the evaluated patterns in the paper. Because of this transformation, the pattern of $Q_{\text{new\_pattern}}$ is indistinguishable from the pattern of $Q_{\text{known\_pattern}}$ from the perspective of the model. Besides the example illustrated above, many other graphs can be converted into one of the patterns in the paper. Since we can perform the conversion in both directions, all these possible patterns are also implicitly included in our used datasets.

In principle, the encoder does also not assume that there are no cycles in the query graph. However, the effect of cycles requires further investigation.

## G    NUMBER OF MESSAGE PASSING STEPS EQUAL TO THE DIAMETER

In Daza & Cochez (2020), the authors find that the best results are achieved when making the number of message passing steps equal to the diameter of the query, defined as the longest shortest path between 2 nodes in it. After these steps, they use the embedding of the target variable as the embedding of the complete graph. Accordingly, we performed additional experiments with this *dynamic query embedding setting*, and with a variant which uses an extra message passing step. From these experiments we concluded that this approach did not lead to better results, which contradicts with the findings in Daza & Cochez (2020).

## H    QUALIFIER IMPACT ON PERFORMANCE

For a more fine-grained analysis of qualifiers impact on query answering performance, for each query pattern we ran an experiment measuring relative performance change in the presence and absence of a given qualifier relation in a query. That is, the main results of StarQE in Table 1 assume all qualifiers are enabled. Then, for each qualifier relation, we remove qualifier pairs containing this relation from all queries in a given pattern and run a model forward pass to obtain new predictions. We run such experiments 5 times for each relation, each metric, and each pattern to get an average value with standard deviations. Finally, for each metric, we sort relations in the ascending order of their performance increase.

Table 4 reports top-3 worst (i.e., first 3 relations in the sorted list) and top-3 best (last 3 relations in the sorted list) qualifier relations that have the most impact along the metrics. For example, `P1686` (`for work`), a common qualifier of relation `award received`, increases Hits@10 performance in 1p-2i queries by a large margin of 62% and MRR by 57% compared to queries without this qualifier. The gains are consistent, although on a smaller scale, in other patterns, too. On the other hand, `P453` (`character role`), a qualifier of `cast member`, seems to be the most confusing qualifier in 2p queries leading to lower scores across all metrics.

We note, however, that on a bigger picture (Fig. 8), where impact for all qualifier relations is visualized, total gains of having qualifiers outweigh negative effects from some of them.

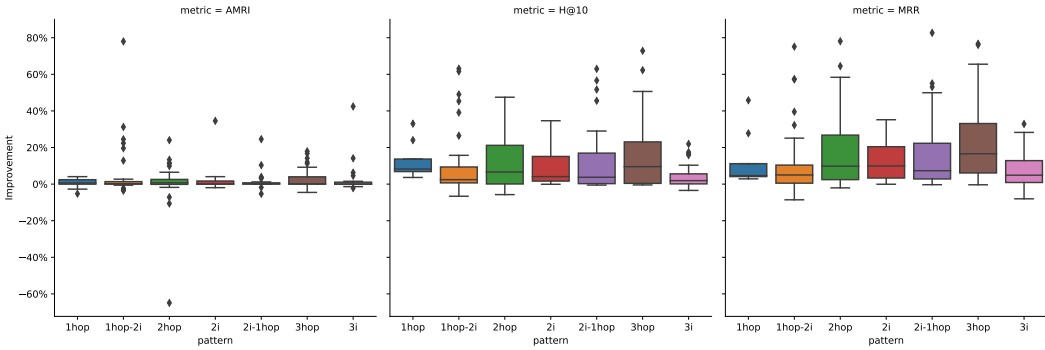

Figure 8: A general overview of qualifiers impact on AMRI, Hits@10, and MRR. In all metrics, higher deltas correspond to better prediction performance. Having qualifiers does, on average, lead to better predictions.

| pattern | relation | AMRI improvement | relation | H@10 improvement | relation | MRR improvement |
|---|---|---|---|---|---|---|
| 1p | P518 | − 5.24 ± 2.09 | P1264 | + 3.64 ± 1.50 | P518 | + 2.88 ± 0.86 |
| | P453 | − 2.75 ± 0.78 | P453 | + 6.00 ± 2.70 | P453 | + 3.78 ± 0.98 |
| | P1264 | − 0.02 ± 0.42 | P518 | + 6.84 ± 4.47 | P17 | + 4.12 ± 1.23 |
| | P805 | + 2.42 ± 2.58 | P1686 | +13.70 ± 1.19 | P1264 | +11.17 ± 2.30 |
| | P17 | + 2.53 ± 3.02 | P1346 | +24.04 ± 3.87 | P1346 | +27.78 ± 4.01 |
| | P1346 | + 4.08 ± 3.30 | P2453 | +33.03 ± 4.26 | P2453 | +45.82 ± 3.48 |
| 2p | P453 | −64.91 ± 40.79 | P459 | − 5.74 ± 2.76 | P453 | − 2.05 ± 5.57 |
| | P2241 | −10.62 ± 50.67 | P453 | − 5.15 ± 12.26 | P3680 | − 0.33 ± 0.55 |
| | P102 | − 7.14 ± 23.57 | P1011 | − 4.55 ± 24.48 | P1552 | − 0.01 ± 0.59 |
| | P805 | +11.56 ± 2.26 | P122 | +45.00 ± 0.00 | P407 | +58.38 ± 7.57 |
| | P531 | +13.35 ± 32.37 | P1686 | +47.00 ± 0.38 | P1310 | +64.49 ± 7.77 |
| | P1686 | +24.04 ± 6.48 | P837 | +47.49 ± 5.83 | P837 | +78.12 ± 1.50 |
| 3p | P102 | − 4.53 ± 15.78 | P155 | − 0.44 ± 0.99 | P3680 | − 0.36 ± 0.57 |
| | P453 | − 4.23 ± 2.75 | P1552 | − 0.16 ± 0.32 | P175 | − 0.30 ± 0.46 |
| | P2241 | − 2.65 ± 12.77 | P937 | + 0.00 ± 0.00 | P1441 | − 0.02 ± 0.04 |
| | P2937 | +14.16 ± 21.84 | P812 | +50.64 ± 11.27 | P1310 | +65.54 ± 7.14 |
| | P92 | +16.77 ± 14.57 | P122 | +62.28 ± 0.00 | P3823 | +76.07 ± 10.71 |
| | P805 | +17.95 ± 2.53 | P3823 | +72.85 ± 42.07 | P837 | +76.71 ± 1.65 |
| 2i | P2715 | − 1.93 ± 5.53 | P291 | − 0.08 ± 0.18 | P3680 | − 0.07 ± 1.16 |
| | P453 | − 0.85 ± 3.90 | P3680 | + 0.00 ± 0.00 | P291 | − 0.06 ± 0.00 |
| | P4100 | − 0.67 ± 2.06 | P2614 | + 0.00 ± 0.00 | P459 | + 0.09 ± 1.11 |
| | P156 | + 3.42 ± 2.93 | P518 | +23.39 ± 4.00 | P1686 | +29.23 ± 1.98 |
| | P17 | + 4.08 ± 5.46 | P1686 | +29.49 ± 2.32 | P1264 | +34.53 ± 16.62 |
| | P805 | +34.56 ± 5.43 | P805 | +34.64 ± 2.82 | P805 | +35.19 ± 1.50 |
| 3i | P453 | − 2.18 ± 4.01 | P2241 | − 3.44 ± 4.47 | P2241 | − 8.03 ± 5.50 |
| | P2241 | − 1.37 ± 5.09 | P4100 | − 0.33 ± 1.73 | P3680 | − 0.01 ± 0.27 |
| | P4100 | − 0.77 ± 1.09 | P1534 | − 0.33 ± 1.13 | P291 | − 0.00 ± 0.00 |
| | P1013 | + 6.28 ± 4.06 | P805 | +16.72 ± 2.81 | P642 | +22.43 ± 15.70 |
| | P805 | +14.13 ± 2.31 | P3831 | +17.49 ± 2.35 | P518 | +28.26 ± 2.68 |
| | P2842 | +42.44 ± 36.23 | P518 | +21.94 ± 1.42 | P1264 | +32.88 ± 25.04 |
| 2i−1p | P2241 | − 5.29 ± 4.07 | P1346 | − 0.53 ± 0.11 | P3680 | − 0.32 ± 0.31 |
| | P459 | − 1.81 ± 7.30 | P2453 | − 0.05 ± 0.11 | P2453 | + 0.03 ± 0.43 |
| | P366 | − 0.01 ± 0.02 | P3680 | + 0.00 ± 0.00 | P1441 | + 0.40 ± 1.20 |
| | P131 | + 3.89 ± 4.81 | P39 | +51.71 ± 36.91 | P1686 | +53.11 ± 0.58 |
| | P805 | +10.37 ± 5.57 | P1686 | +56.63 ± 0.73 | P39 | +55.05 ± 23.43 |
| | P1686 | +24.58 ± 6.44 | P837 | +62.98 ± 27.55 | P837 | +82.67 ± 2.60 |
| 1p−2i | P2241 | − 3.56 ± 8.66 | P459 | − 6.63 ± 14.51 | P2241 | − 8.57 ± 7.45 |
| | P17 | − 3.30 ± 4.91 | P407 | − 6.34 ± 14.91 | P1039 | − 3.33 ± 6.94 |
| | P453 | − 2.66 ± 5.26 | P2241 | − 3.13 ± 3.40 | P92 | − 0.70 ± 2.65 |
| | P1013 | +24.53 ± 21.48 | P102 | +49.04 ± 12.18 | P1686 | +57.30 ± 1.64 |
| | P654 | +31.24 ± 46.20 | P1686 | +61.66 ± 2.20 | P805 | +57.36 ± 0.98 |
| | P805 | +77.99 ± 10.23 | P805 | +63.10 ± 0.38 | P837 | +75.11 ± 3.91 |

Table 4: Top 3 worst and top 3 best impacting qualifier relations per pattern. In all metrics (AMRI, H@10, MRR), positive value corresponds to better predictions. For some metrics, and given patterns we only see improvements in cases where qualifiers are included. The qualifier relation P453 (*specific role played or filled by subject – as a "cast member" or "voice actor"*) seems to confuse the model the most. On the other hand P805 (*referring to an item that describes the relation identified in the statement*) often leads to the biggest improvements in the metrics. All relation names are clickable links to their Wikidata pages.

# I    THE ORACLE SETUP

There are several QE methods we could compare our work with. However, these methods are only able to answer queries utilising triple and not qualifier information. Besides, we see that new methods are introduced regularly, each outperforming the previous method. Hence, we employ an Oracle approach to compute the upper bound that triple only QE models can achieve on WD50K-QE. This Oracle is simulating perfect link prediction and ranking capabilities but does not have access to qualifier information.

To achieve this, the Oracle system has access to training, validation, as well as *test* data. Given the access to the totality of the data, the system will return an optimal ordering, i.e., one that maximizes our reported metrics. However, since the system cannot process qualifier information, the ordering of the result set can only be based on the entities and relations in the query. This means that queries that are the same when ignoring the qualifier information, will get the same list of answers. In effect, the oracle will, despite its perfect information for a setting without qualifiers, not answer perfectly. Using this setting we can hence investigate how much difference qualifier information makes for our queries.

Finally, note that because multiple ordering can be considered optimal from the information the Oracle has available, we report the expected value for these metrics.

# J    DETAILED RESULTS

We provide our chosen hyper-parameters after performing hyper-parameter optimisation in Table 6 and detailed results including standard deviation across five runs with different random seeds in Tables 7 and 8. We report Hits@k for $k = 1, 3, 10$, mean reciprocal rank (MRR), and adjusted mean rank index (AMRI) (Berrendorf et al., 2020). For all metrics, larger values indicate better performance. We highlight the best result per column and metric in **bold font**.

For the Hits@k metric, we observe StarQE performing best across patterns, except for the most simple ones, `1p` (equivalent to plain link prediction), and `3i`. Standard deviations are usually small (around 1% point). The base triples baseline sometimes exhibits larger variances across multiple random seeds. For MRR we make similar observations as for Hits@k, which is intuitive since it can be seen as a soft version of H@k.

For AMRI, the picture differs slightly. AMRI is a linear transformation of the mean rank, which normalizes the scores such that 0 corresponds to the performance of a random scoring model and 100% to the perfect score. Besides, AMRI preserves the linearity of mean rank, i.e., it is equally influenced by improvements on any rank, not just at the top of the ranking. Across the board, we observe values far beyond 50%, usually exceeding 90%. The general tendency of some patterns being more difficult than others persists and is coherent with the other metrics. Comparing different models, we can see that StarQE falls behind, e.g., reification and more simple baselines such as zero-layers, which do not use the graph structure. Since this behavior was not observed for the metrics more focused on the top positions in ranking, the decrease in performance in this metric has to be caused by entities that received large ranks in the first place.

Zooming in on the Oracle setup, we make several observations. First, it becomes increasingly harder for the Oracle to produce good results for smaller values of $k$ in the Hits@k metrics. Since this is an upper bound to the Base triple setup, also that model shows similar behavior. The observation that the intersection queries can still perform well is also visible in the more detailed results.

The very high AMRI for the Oracle is also expected. Since this model is designed as the best possible ranking model, it will place all correct answers for the query (while ignoring qualifiers) at the top of the ranking. This set of answers is a super set of the correct answers to the qualified query (see Appendix D). Now, that set of answers is nearly always very small compared to the 50K candidate entities in our dataset. So, relatively speaking the correct answers are still nearly always ranked high, and hence we obtain a high AMRI score.

# K    EVALUATING FAITHFULNESS

Following the idea of EmQL (Sun et al., 2020), we evaluate *faithfulness* of our hyper-relational approach by evaluating its performance on the training set, that is, an ability to correctly answer already seen queries. Evaluation on the training set is a suitable proxy for faithfulness since even the union of training, validation and test queries (as done in the original work) is highly incomplete considering the whole background graph (be in Freebase or Wikidata). To this end, we evaluate the model trained in two regimes: *StarQE-like* trained on all query types and *MPQE-like* trained in the hardest setting on only `1p` link prediction queries. The results presented in Table 5 show that the *StarQE-like* model exhibits faithfulness saturating performance metrics to almost perfect results. As expected, training the model only on one query type inhibits faithfulness qualities.

Table 5: Full results for the **faithfulness** experiments, including standard deviation across five runs with different random seeds. We report Hits@k for $k = 1, 3, 10$, mean reciprocal rank (MRR), and adjusted mean rank index (AMRI) Berrendorf et al. (2020). For all metrics, larger values indicate better performance.

| Pattern | 1p | 2p | 3p | 2i | 3i | 2i-1p | 1p-2i |
|---|---|---|---|---|---|---|---|
| | | | | Hits@1 (%) | | | |
| StarQE-like | $74.27 \pm 5.30$ | $94.58 \pm 1.81$ | $89.93 \pm 2.29$ | $94.29 \pm 1.67$ | $99.34 \pm 0.24$ | $96.02 \pm 1.45$ | $99.05 \pm 0.46$ |
| MPQE-like | $90.43 \pm 2.10$ | $8.00 \pm 0.85$ | $19.99 \pm 2.05$ | $87.88 \pm 1.53$ | $92.44 \pm 1.56$ | $13.68 \pm 0.97$ | $65.51 \pm 2.60$ |
| | | | | Hits@3 (%) | | | |
| StarQE-like | $84.06 \pm 4.64$ | $98.12 \pm 0.49$ | $96.56 \pm 0.89$ | $97.48 \pm 1.15$ | $99.81 \pm 0.11$ | $98.78 \pm 0.51$ | $99.84 \pm 0.09$ |
| MPQE-like | $96.18 \pm 1.10$ | $14.19 \pm 1.18$ | $32.12 \pm 2.59$ | $97.17 \pm 0.62$ | $99.46 \pm 0.16$ | $18.64 \pm 1.43$ | $85.42 \pm 1.24$ |
| | | | | Hits@10 (%) | | | |
| StarQE-like | $90.94 \pm 4.18$ | $99.39 \pm 0.16$ | $98.98 \pm 0.38$ | $99.18 \pm 0.67$ | $99.96 \pm 0.04$ | $99.64 \pm 0.20$ | $99.97 \pm 0.03$ |
| MPQE-like | $98.75 \pm 0.41$ | $22.62 \pm 1.56$ | $47.09 \pm 3.14$ | $99.46 \pm 0.18$ | $99.94 \pm 0.02$ | $23.98 \pm 1.70$ | $94.90 \pm 0.35$ |
| | | | | MRR (%) | | | |
| StarQE-like | $80.23 \pm 4.85$ | $96.51 \pm 1.09$ | $93.50 \pm 1.51$ | $96.09 \pm 1.32$ | $99.59 \pm 0.17$ | $97.47 \pm 0.95$ | $99.45 \pm 0.27$ |
| MPQE-like | $93.58 \pm 1.52$ | $12.88 \pm 1.06$ | $28.88 \pm 2.36$ | $92.70 \pm 0.94$ | $95.93 \pm 0.81$ | $17.36 \pm 1.23$ | $76.54 \pm 1.74$ |
| | | | | AMRI (%) | | | |
| StarQE-like | $99.97 \pm 0.03$ | $99.99 \pm 0.00$ | $99.99 \pm 0.00$ | $100.00 \pm 0.00$ | $100.00 \pm 0.00$ | $100.00 \pm 0.00$ | $100.00 \pm 0.00$ |
| MPQE-like | $100.00 \pm 0.00$ | $91.77 \pm 2.29$ | $94.64 \pm 2.59$ | $100.00 \pm 0.00$ | $100.00 \pm 0.00$ | $93.97 \pm 2.09$ | $99.99 \pm 0.00$ |

Table 6: Best hyper-parameter as chosen after hyper-parameter optimization

| Experiment | StarQE | Reification | Base Triple | Zero Layers | MPQE-like | MPQE-like + Reif | EmQL-like | Q2B-like |
|---|---|---|---|---|---|---|---|---|
| activation | leakyrelu | relu | relu | relu | prelu | relu | leakyrelu | leakyrelu |
| optimiser | adam | adam | adam | adam | adam | adam | adam | adam |
| learning-rate | 0.0007741 | 0.003768 | 0.007253 | 0.0008733 | 0.0005256 | 0.0001414 | 0.007253 | 0.002075 |
| batch-size | 64 | 32 | 32 | 64 | 128 | 32 | 128 | 32 |
| graph-pooling | targetpooling | sum | sum | sum | targetpooling | targetpooling | targetpooling | sum |
| message-weighting | attention | attention | attention | symmetric | attention | symmetric | symmetric | attention |
| similarity | dotproduct | negativepowernorm | negativepowernorm | negativepowernorm | dotproduct | dotproduct | dotproduct | negativepowernorm |
| num-layers | 3 | 2 | 2 | 2 | 3 | 3 | 3 | 3 |
| use-bias | True | True | True | False | False | False | False | True |
| embedding-dim | 192 | 224 | 224 | 160 | 128 | 96 | 256 | 224 |
| dropout | 0.5 | 0.3 | 0.3 | 0.3 | 0.5 | 0.2 | 0.3 | 0.1 |
| composition | multiplication | multiplication | multiplication | multiplication | multiplication | multiplication | multiplication | multiplication |

Table 7: Full results for baseline experiments, including standard deviation across five runs with different random seeds. We report Hits@k for $k = 1, 3, 10$, mean reciprocal rank (MRR), and adjusted mean rank index (AMRI) Berrendorf et al. (2020). For all metrics, larger values indicate better performance. We highlight the best result per column and metric (excluding the Oracle) in **bold font.**

| Pattern | 1p | 2p | 3p | 2i | 3i | 2i–1p | 1p–2i |
|---|---|---|---|---|---|---|---|
| **Hits@1 (%)** | | | | | | | |
| StarQE | $20.91 \pm 1.11$ | $\mathbf{39.98 \pm 0.27}$ | $\mathbf{45.85 \pm 0.81}$ | $\mathbf{55.11 \pm 1.26}$ | $78.58 \pm 1.44$ | $\mathbf{51.77 \pm 0.73}$ | $\mathbf{65.77 \pm 1.87}$ |
| Base Triple | $11.62 \pm 0.68$ | $2.97 \pm 0.21$ | $7.57 \pm 1.59$ | $36.87 \pm 2.22$ | $64.72 \pm 6.97$ | $3.33 \pm 0.33$ | $13.00 \pm 1.07$ |
| Reification | $\mathbf{24.26 \pm 0.88}$ | $38.82 \pm 0.41$ | $42.80 \pm 0.50$ | $49.67 \pm 1.86$ | $\mathbf{78.93 \pm 3.20}$ | $42.39 \pm 0.68$ | $55.05 \pm 1.11$ |
| Zero Layers | $18.30 \pm 0.31$ | $12.92 \pm 0.25$ | $11.53 \pm 0.13$ | $40.35 \pm 0.82$ | $74.27 \pm 0.32$ | $18.17 \pm 0.25$ | $19.51 \pm 0.22$ |
| Oracle | $37.87$ | $7.19$ | $13.86$ | $75.67$ | $94.07$ | $7.02$ | $32.61$ |
| **Hits@3 (%)** | | | | | | | |
| StarQE | $35.58 \pm 0.82$ | $\mathbf{46.41 \pm 0.32}$ | $\mathbf{57.34 \pm 0.58}$ | $67.98 \pm 0.93$ | $87.56 \pm 0.74$ | $\mathbf{56.89 \pm 0.47}$ | $\mathbf{75.35 \pm 1.02}$ |
| Base Triple | $25.54 \pm 0.28$ | $6.38 \pm 0.32$ | $15.44 \pm 1.56$ | $53.45 \pm 2.60$ | $81.55 \pm 4.70$ | $7.40 \pm 0.67$ | $23.56 \pm 1.49$ |
| Reification | $\mathbf{40.12 \pm 1.37}$ | $45.58 \pm 0.74$ | $55.45 \pm 0.88$ | $\mathbf{68.31 \pm 0.77}$ | $\mathbf{89.84 \pm 1.90}$ | $51.33 \pm 0.37$ | $70.26 \pm 0.83$ |
| Zero Layers | $31.98 \pm 0.23$ | $21.34 \pm 0.34$ | $25.24 \pm 0.27$ | $55.83 \pm 0.42$ | $85.43 \pm 0.37$ | $25.75 \pm 0.26$ | $35.67 \pm 0.28$ |
| Oracle | $61.65$ | $13.06$ | $24.74$ | $88.42$ | $98.36$ | $15.20$ | $53.94$ |
| **Hits@10 (%)** | | | | | | | |
| StarQE | $51.72 \pm 0.66$ | $\mathbf{51.20 \pm 0.44}$ | $\mathbf{65.50 \pm 0.39}$ | $77.78 \pm 0.53$ | $92.64 \pm 0.65$ | $\mathbf{61.81 \pm 0.37}$ | $\mathbf{81.60 \pm 0.60}$ |
| Base Triple | $45.04 \pm 1.18$ | $12.76 \pm 0.92$ | $24.66 \pm 2.15$ | $69.74 \pm 1.43$ | $91.74 \pm 1.83$ | $16.77 \pm 1.44$ | $40.67 \pm 1.31$ |
| Reification | $\mathbf{55.17 \pm 1.00}$ | $50.86 \pm 0.39$ | $63.65 \pm 0.60$ | $\mathbf{81.25 \pm 0.45}$ | $\mathbf{95.31 \pm 0.78}$ | $61.05 \pm 0.92$ | $80.49 \pm 0.81$ |
| Zero Layers | $44.93 \pm 0.35$ | $29.94 \pm 0.35$ | $38.45 \pm 0.18$ | $67.79 \pm 0.42$ | $90.66 \pm 0.20$ | $35.48 \pm 0.31$ | $47.85 \pm 0.43$ |
| Oracle | $81.03$ | $24.11$ | $38.47$ | $95.54$ | $99.67$ | $32.74$ | $76.96$ |
| **MRR (%)** | | | | | | | |
| StarQE | $30.98 \pm 0.91$ | $\mathbf{44.13 \pm 0.28}$ | $\mathbf{52.96 \pm 0.61}$ | $\mathbf{63.14 \pm 0.97}$ | $83.78 \pm 1.00$ | $\mathbf{55.20 \pm 0.52}$ | $\mathbf{71.52 \pm 1.37}$ |
| Base Triple | $22.25 \pm 0.25$ | $6.73 \pm 0.28$ | $14.05 \pm 1.72$ | $48.01 \pm 2.13$ | $74.52 \pm 5.35$ | $8.16 \pm 0.62$ | $22.23 \pm 1.21$ |
| Reification | $\mathbf{34.78 \pm 1.07}$ | $43.42 \pm 0.48$ | $50.77 \pm 0.54$ | $61.01 \pm 1.17$ | $\mathbf{85.17 \pm 2.33}$ | $49.09 \pm 0.63$ | $64.21 \pm 0.86$ |
| Zero Layers | $27.55 \pm 0.19$ | $19.10 \pm 0.27$ | $21.25 \pm 0.14$ | $50.27 \pm 0.57$ | $80.62 \pm 0.28$ | $24.49 \pm 0.22$ | $30.04 \pm 0.21$ |
| Oracle | $52.64$ | $13.44$ | $22.65$ | $83.00$ | $96.34$ | $15.84$ | $47.00$ |
| **AMRI (%)** | | | | | | | |
| StarQE | $78.44 \pm 1.78$ | $68.11 \pm 2.52$ | $74.62 \pm 2.94$ | $93.63 \pm 0.81$ | $98.65 \pm 0.33$ | $73.14 \pm 2.39$ | $88.76 \pm 0.61$ |
| Base Triple | $88.14 \pm 0.21$ | $81.25 \pm 0.46$ | $95.92 \pm 0.11$ | $99.18 \pm 0.12$ | $99.95 \pm 0.01$ | $83.44 \pm 0.95$ | $98.91 \pm 0.11$ |
| Reification | $\mathbf{88.81 \pm 0.47}$ | $86.63 \pm 1.41$ | $\mathbf{96.20 \pm 0.28}$ | $\mathbf{99.24 \pm 0.10}$ | $\mathbf{99.96 \pm 0.01}$ | $85.46 \pm 0.82$ | $\mathbf{99.10 \pm 0.12}$ |
| Zero Layers | $88.03 \pm 0.22$ | $\mathbf{90.56 \pm 0.47}$ | $95.79 \pm 0.20$ | $96.34 \pm 0.30$ | $98.37 \pm 0.13$ | $\mathbf{86.18 \pm 0.60}$ | $98.55 \pm 0.10$ |
| Oracle | $99.97$ | $99.77$ | $99.83$ | $99.99$ | $100.00$ | $99.88$ | $99.97$ |

Table 8: Full results for the generalization experiments, including standard deviation across five runs with different random seeds. We report Hits@k for $k = 1, 3, 10$, mean reciprocal rank (MRR), and adjusted mean rank index (AMRI) Berrendorf et al. (2020). For all metrics, larger values indicate better performance. We highlight the best result per column and metric in **bold font**.

| Pattern | 1p | 2p | 3p | 2i | 3i | 2i-1p | 1p-2i |
|---|---|---|---|---|---|---|---|
| **Hits@1 (%)** | | | | | | | |
| StarE-like | 20.91 ± 1.11 | **39.98 ± 0.27** | **45.85 ± 0.81** | **55.11 ± 1.26** | 78.58 ± 1.44 | **51.77 ± 0.73** | **65.77 ± 1.87** |
| Q2B-like | 21.63 ± 0.70 | 37.15 ± 0.65 | 43.82 ± 0.98 | 52.49 ± 1.45 | 77.36 ± 1.46 | 37.59 ± 2.88 | 59.74 ± 3.29 |
| emQL-like | 23.06 ± 1.12 | 6.78 ± 2.24 | 19.86 ± 4.99 | 53.36 ± 3.46 | **81.73 ± 3.63** | 2.17 ± 0.87 | 48.54 ± 5.09 |
| MPQE-like | 16.53 ± 0.48 | 3.81 ± 0.38 | 12.70 ± 1.59 | 41.01 ± 1.49 | 60.67 ± 2.23 | 6.84 ± 0.58 | 28.76 ± 1.40 |
| MPQE-like + Reif | **25.52 ± 0.36** | 3.24 ± 0.62 | 10.52 ± 2.09 | 41.36 ± 0.59 | 63.93 ± 1.36 | 5.59 ± 0.72 | 18.75 ± 1.55 |
| **Hits@3 (%)** | | | | | | | |
| StarE-like | 35.58 ± 0.82 | **46.41 ± 0.32** | **57.34 ± 0.58** | 67.98 ± 0.93 | 87.56 ± 0.74 | **56.89 ± 0.47** | **75.35 ± 1.02** |
| Q2B-like | 38.79 ± 0.92 | 43.75 ± 0.98 | 55.41 ± 1.21 | 67.02 ± 1.24 | 87.97 ± 1.23 | 46.99 ± 3.04 | 71.20 ± 2.45 |
| emQL-like | 36.14 ± 1.11 | 10.73 ± 3.23 | 30.47 ± 6.37 | 66.56 ± 3.18 | 89.91 ± 2.87 | 3.80 ± 1.05 | 55.96 ± 5.31 |
| MPQE-like | 30.82 ± 0.75 | 7.15 ± 0.65 | 21.48 ± 2.58 | 66.87 ± 1.21 | 86.17 ± 1.45 | 10.08 ± 1.08 | 44.88 ± 1.30 |
| MPQE-like + Reif | **41.26 ± 0.65** | 6.34 ± 0.82 | 18.25 ± 2.69 | **68.14 ± 0.97** | **90.07 ± 0.58** | 8.56 ± 0.59 | 32.31 ± 2.60 |
| **Hits@10 (%)** | | | | | | | |
| StarE-like | 51.72 ± 0.66 | **51.20 ± 0.44** | 65.50 ± 0.39 | 77.78 ± 0.53 | 92.64 ± 0.65 | **61.81 ± 0.37** | **81.60 ± 0.60** |
| Q2B-like | 55.44 ± 1.35 | 51.10 ± 1.40 | **66.39 ± 1.72** | 78.79 ± 1.04 | 94.20 ± 0.86 | 57.49 ± 2.26 | 80.49 ± 1.92 |
| emQL-like | 50.10 ± 1.34 | 16.45 ± 3.56 | 44.36 ± 4.96 | 75.86 ± 2.45 | 93.55 ± 1.93 | 6.79 ± 1.13 | 62.80 ± 5.26 |
| MPQE-like | 48.48 ± 0.59 | 12.57 ± 0.96 | 34.19 ± 3.78 | 83.04 ± 1.01 | 96.32 ± 0.77 | 14.75 ± 1.42 | 61.02 ± 0.70 |
| MPQE-like + Reif | **58.43 ± 0.29** | 12.02 ± 1.12 | 31.14 ± 2.70 | **83.77 ± 0.47** | **97.22 ± 0.14** | 13.50 ± 0.81 | 50.92 ± 3.44 |
| **MRR (%)** | | | | | | | |
| StarE-like | 30.98 ± 0.91 | **44.13 ± 0.28** | **52.96 ± 0.61** | **63.14 ± 0.97** | 83.78 ± 1.00 | **55.20 ± 0.52** | **71.52 ± 1.37** |
| Q2B-like | 33.04 ± 0.84 | 41.99 ± 0.88 | 51.71 ± 1.18 | 61.72 ± 1.18 | 83.49 ± 1.24 | 44.24 ± 2.73 | 67.04 ± 2.70 |
| emQL-like | 32.01 ± 1.00 | 10.09 ± 2.70 | 27.94 ± 5.11 | 61.45 ± 3.13 | **86.28 ± 3.11** | 3.73 ± 0.94 | 53.58 ± 5.03 |
| MPQE-like | 26.83 ± 0.53 | 6.79 ± 0.56 | 19.72 ± 2.23 | 56.16 ± 1.30 | 74.35 ± 1.70 | 9.62 ± 0.85 | 39.81 ± 1.20 |
| MPQE-like + Reif | **36.36 ± 0.38** | 6.12 ± 0.74 | 17.11 ± 2.27 | 56.81 ± 0.69 | 77.29 ± 0.86 | 8.32 ± 0.52 | 29.25 ± 2.05 |
| **AMRI (%)** | | | | | | | |
| StarE-like | 78.44 ± 1.78 | 68.11 ± 2.52 | 74.62 ± 2.94 | 93.63 ± 0.81 | 98.65 ± 0.33 | 73.14 ± 2.39 | 88.76 ± 0.61 |
| Q2B-like | 88.69 ± 0.45 | **87.17 ± 1.48** | **95.60 ± 0.58** | 99.01 ± 0.20 | 99.93 ± 0.03 | 83.24 ± 1.08 | 98.49 ± 0.40 |
| emQL-like | 73.20 ± 11.74 | 51.74 ± 13.25 | 74.93 ± 10.70 | 90.68 ± 5.62 | 97.79 ± 1.60 | 37.80 ± 22.89 | 82.59 ± 8.45 |
| MPQE-like | 93.77 ± 0.09 | 83.25 ± 2.61 | 92.00 ± 3.04 | 99.49 ± 0.05 | 99.96 ± 0.01 | 85.61 ± 2.86 | 99.04 ± 0.16 |
| MPQE-like + Reif | **95.68 ± 0.09** | 82.93 ± 1.83 | 92.08 ± 1.43 | **99.57 ± 0.02** | **99.97 ± 0.00** | **86.54 ± 1.56** | **99.25 ± 0.06** |

