# OpenReview forum: "Query Embedding on Hyper-Relational Knowledge Graphs"
_ICLR.cc/2022/Conference — ICLR 2022 Poster_

### Official Review · Reviewer_Bute · 2021-11-02

**Correctness:** 4
**Technical Novelty And Significance:** 3
**Empirical Novelty And Significance:** 3
**Recommendation:** 8
**Confidence:** 4

**Main Review:**

Strengths.
- S1. They proposed the first query embedding approach for a graph with qualifiers.
- S2. The proposed approach is compared with three baselines including the reification baseline.
- S3. A new evaluation dataset is proposed.

Weaknesses.
- W1. Some of important information had to go to Appendix due to space. For example, I think the best hyperparameters used in the experiment can stay in the main paper.
- W2. The proposed approach performs worse than the reification baseline when the problem is easier and the answer can be narrow down with traditional conjunctions.
- W3. Another baseline of embedding the entire edge-qualifiers pair as a unit rather than representing the qualifiers as separate nodes is not considered. It can be predicted that this baseline is not suitable for a large number of possible qualifier values, but can be easily applicable to the suggested example by considering three atomic relations instead of qualifiers: educatedAt-BSc, educatedAt-MS, educatedAT-PhD.

**Summary Of The Paper:**

This paper proposes a novel problem to embed query graphs with edge qualifiers to query a hyper-relational knowledge graph, and a solution extending an existing approach for a non-hyper-relational knowledge graph. Due to the lack of evaluation datasets, they propose to use a hyper-relational knowledge graph extracted from Wikidata. They also consider three baselines that can show the characteristics of the proposed approach, including reification baseline that transforms qualifiers to nodes. The overall performance shows the proposed approach shows benefit over the baselines, and usefulness when edge qualifiers are available.

**Summary Of The Review:**

I think this is a solid piece of work extending the problem space solved by modern embedding techniques. The presentation is mostly solid, and easy to read and understand. There is some very minor presentation issues that can be improved such as adding the best hyperparameters in the main paper, or adding the missing node color in the legend in Figure 4 or maybe using "Oracle-" instead of just "Oracle" which can be a bit misleading. But otherwise, this paper is well written with a clear goal and contributions, and evaluations. Since in case of limited set of qualifier values, the edge+qualifiers can be easily represented and embedded as altogether, it would be interesting to see the comparison with such a baseline. But I don't think this is necessary to show the value of this paper.

---

> ### Author Response · Authors · 2021-11-17
> **Response to Bute**
>
>
> We would like to thank the Reviewer for his feedback. The acknowledgement of the novelty in our approach as well as the compelling experimental program is gratifying.
>
> >W1. Some of important information had to go to Appendix due to space. For example, I think the best hyperparameters used in the experiment can stay in the main paper.
>
> We moved such large tables in the Appendix due to space limitations. Given the multiple different setups, it is not feasible to include them all in the main part of the revised version.
>
>
> >W2. The proposed approach performs worse than the reification baseline when the problem is easier and the answer can be narrow down with traditional conjunctions.
>
> The fact that reification does show similar (or better) performance to the qualifier embedding case, shows that our model is rather __robust__ to the representation of hyper-relational facts and does not lose the _semantic_ information encoded in reified or hyper-relational facts. That is, we would rather treat this fact as a positive signal.
> Note that reification is not a baseline which we aim to beat but rather, a different way of representating hyper-relational information that comes with specific costs and benefits (e.g. the change in query structure, different structural properties of the query graph, etc.).
>
> That being said, however, the difference between hyper-relational and reification models where reification performs better is quite marginal - often withing the standard deviation. We refer the Reviewer to our Appendix and, specifically, Table 7 where we provide standard deviations accross 5 runs for different seeds.
>
>
>
> >W3. Another baseline of embedding the entire edge-qualifiers pair as a unit rather than representing the qualifiers as separate nodes is not considered. It can be predicted that this baseline is not suitable for a large number of possible qualifier values, but can be easily applicable to the suggested example by considering three atomic relations instead of qualifiers: educatedAt-BSc, educatedAt-MS, educatedAT-PhD.
>
>
> This is a very nice idea, which we started to work out. First, as you correctly indicate, this approach might lead to issues in theory. Specifically, each relation can have al large number of (qualifier relation, qualifier value) pairs attached to it (in our formulation up to $2^{(\mathcal{R}\times\mathcal{E})}$). Regarding these as a unit together with the relation type, would in theory lead to a very large number of new relations ($\mathcal{R}\times2^{(\mathcal{R}\times\mathcal{E})}$).
> In real data, however, one would never combine every qualifier relation, with every qualifier value, as the semantic meaning becomes non-sensical. Besides, some relation types would only be used for qualifier relations, while others would only be used as a main relation. Hence, the question is what the situation is like in our dataset.
> What we observe in our overall dataset is that the total number of *unique* (qualifier relation, qualifier value) pairs is 13.7K on a total of about 50K occuring pairs. A bit over 1000 of these unique pairs do not occur at all in our train and validation set. Further, we notice that in our training set the distribution of these pairs is long-tailed. About 8K occur only once, 3.5K twice, 800 three times; and the remaining 1.4 K occur more often. What we are counting here is the number of (qualifier relation, qualifier value) pairs, meaning that if we would combine these with the main relation to form a unit, we would get even fewer occurences for each.
> This leads to certain problems. First of all, no system using this technique could learn a representation for pairs which do not occur in our training set. Second, having only one or very few examples for a relation, is insufficient for a graph neural network to be trained properly. One approach could rely on weight sharing to aleviate this, but then we are essentially back to the proposed architecture.
> Therefore, a compositional encoder that builds qualifier representations from basic atoms (entities and relations) is a much more preferable option.
>
> We do wonder whether it would be be useful to include these characteristics of our dataset in the appendix of the paper?

---

> > ### Comment · Reviewer_Bute · 2021-11-20
> > **Acknowledging the comments**
> >
> > I read the authors' response, and the answers are just as expected. As I stated before, I would consider none of these as a basis to reject the paper. The paper will be only further improved by including some of the discussions here with a concise language. I agree that the proposed method is a simple combination of two existing approaches, but it is very well done, and well justified; the paper discussed why they chose these two among so many approaches out there and their combinations, and did so in a very limited space. Maybe one way to address the concern is using some other options as baselines to further justify the combination, if the paper ever gets rejected. In short, based on the discussions so far, I am keeping my recommendation to accept the paper.

---

### Official Review · Reviewer_GQAR · 2021-11-02

**Correctness:** 3
**Technical Novelty And Significance:** 2
**Empirical Novelty And Significance:** 2
**Recommendation:** 5
**Confidence:** 2

**Main Review:**

The challenge of multi-hop logical reasoning is not clearly stated. I suggest that the author should discuss the challenge in detail.

In the MODELDESCRIPTION section, the author should first explain the challenges of solving the problem and how the proposed method is solved.I suggest the author introduce it in detail.

In the introduction of related work, the author did not analyze the advantages and disadvantages, and did not discuss the proposed method, which could not prove the innovation of the proposed method

There is little description of Query Embedding in this paper.Suggeest the author to add more space to introduce it.

The abstract of the experimental results is not specific enough.

There is only few references in the past 3 years. It is recommended to ensure the latestness of the literature when investigating related work, otherwise it is difficult to persuade the work to be novel.

**Summary Of The Paper:**

This paper studies how to embed and answer hyper-relational conjunctive queries based on recent advancements in Graph Neural Networks and query embedding techniques. And propose a method to answer such queries and demonstrate in their experiments that qualifiers improve QA on a diverse set of query patterns. This paper has a certain novelty.

**Summary Of The Review:**

This paper propose a method to answer such queries and demonstrate in their experiments that qualifiers improve QA on a diverse set of query patterns. The logic is clear. The argument is complete and the grammatical expression is very standard. However, in the process of demonstration, there is a lack of description of the challenge and thinking about how to solve the problem. I think the authors needs to improve the writing.

---

> ### Author Response · Authors · 2021-11-17
> **Response to GQAR**
>
>
> We would like to thank the Reviewer for the recognition of the novelty and clarity of our approach as well as the completeness of our argumentation. We will address the reviewers detailed comments below.
>
> > The challenge of multi-hop logical reasoning is not clearly stated. I suggest that the author should discuss the challenge in detail.
>
> We agree with the Reviewer in that we would have liked to further elaborate on this task. However, due to space limitations, we did not include extensive details about the challenge of multi-hop reasoning, and instead, we opted to include references to prominent and relevant papers that introduced the task and thus describe it with more clarity and in more detail.
>
> >In the MODELDESCRIPTION section, the author should first explain the challenges of solving the problem and how the proposed method is solved.I suggest the author introduce it in detail.
>
> We opted to describe the problem setting in the introduction (and more formally in Section 3), and outlined what is missing in related work in Section 2, which can be briefly summarized as the missing ability to utilize hyper-relational information for query embedding (cf. last paragraph of Section 2). In the model description section, we thus concentrate on describing our proposed method.
>
>
>
> > In the introduction of related work, the author did not analyze the advantages and disadvantages, and did not discuss the proposed method, which could not prove the innovation of the proposed method
>
> The principle weakness of all approaches listed in our Related Work section, lies in the fact that none of the approaches can process hyper-relational information, which is a contribution of our work. We would also point the Reviewer to the last paragraph of our Query Embedding - Related Work section where we emphasize  that. Regarding existing limitations of our approach, they are described in Section 6 - Limitations & Future Work.
>
> > There is little description of Query Embedding in this paper.Suggeest the author to add more space to introduce it.
>
> Due to space limitations, we provide references to the papers that introduced Query Embedding as a task and hence provide much more detail (see, e.g., [1]). In our work, we still try to briefly describe / recapitulate the gist of the problem (cf. Introduction & formalization in Section 3, Dataset description in Section 5.1 and Appendix B).
>
>
> > The abstract of the experimental results is not specific enough.
>
> We are not sure what the Reviewer refers to, and would kindly ask for further clarification?
>
>
> > There is only few references in the past 3 years. It is recommended to ensure the latestness of the literature when investigating related work, otherwise it is difficult to persuade the work to be novel.
>
>
> The first paper introducing the query embedding task by Hamilton et al [1] came out in 2018. A flurry of approaches improving certain components of the original 2018 work started to appear after that including major advancements like Query2Box, BetaE, MPQE and CQD. We are relatively certain to have covered the main body of work, but there may be something we missed. In that case, we would be glad to be pointed towards missing work(s).
>
>
> [1] Hamilton et al. "Embedding logical queries on knowledge graphs." NeurIPS 2018

---

### Official Review · Reviewer_iDXR · 2021-11-02

**Correctness:** 3
**Technical Novelty And Significance:** 2
**Empirical Novelty And Significance:** 3
**Recommendation:** 5
**Confidence:** 4

**Main Review:**

This paper proposed StarQE for question answering over hyper-relational knowledge graphs. Overall, the motivation is strong, and the method is technically sound.
#####Pros#####
1. The task is well-motivated and interesting. It can deal with the question-answering task with qualifier pairs (Yet, some contextual information is not allowed).
2. The proposed model is simple but performs well on certain types of queries. And it has a certain generalization capability.
3. A new dataset is constructed in order to verify the model efficacy.
4. The paper is well-written and easy to follow.

#####Cons#####
1. The experiments could be more convincing by using more datasets. Currently, only one dataset is used. It is not sure that if this method can also be used on other datasets or domains. Also, since the model can handle triple-only scenarios, I suggest the authors also consider reporting the performances on Freebase and Nell as done in query2box, as such, we can see whether or not this model can deal with more basic cases.
2. The novelty is limited. The model is just a combination of two existing approaches.
3. It cannot handle queries about numbers, text, and time. While some existing methods [a][b][c] can handle such information in knowledge graph embeddings.  Those important references are missing and should be used as baselines especially on 1p queries.
4. Important reasoning operators such as negation, disjunctions are not considered, making the approach less attractive. I guess we can draw some insights from betaE for this type of operator.
5. I suggest the authors add some baselines in Table 2 so that we can know the differences between StarQE and baselines in generalization capability.

#####Question#####
Can the model benefit from nonlinear layers (transformations)?

[a] Knowledge Graph Embedding with Numeric Attributes of Entities, ACL 2018
[b] Incorporating Literals into Knowledge Graph Embeddings, ISWC 2019
[c] Beyond Triplets: Hyper-Relational Knowledge Graph Embedding for Link Prediction, WWW 2020


**Summary Of The Paper:**

This paper studies the multi-hop logical reasoning problem with hyper-relational knowledge graphs. In hyper relational knowledge graphs, edges are associated with some key-value pairs used for describing the contextual information. The goal of this paper is to incorporate such information in order to provide more fine-grained question answering. Specifically, StarQE, a combination of MPQEand StarE, is proposed to achieve this goal. The authors constructed a new dataset, WD50K-QE, and the experimental results show that StarQE can effectively model the qualifiers in the dataset.

**Summary Of The Review:**

The current version of this paper is marginally below the acceptance threshold. I believe some improvements can be made to strengthen the paper.

---

> ### Author Response · Authors · 2021-11-17
> **Response iDXR Part I**
>
> We would like to thank the Reviewer for their comments and for highlighting the technical soundness and convincing motivation behind our work.
>
> >The experiments could be more convincing by using more datasets. Currently, only one dataset is used. It is not sure that if this method can also be used on other datasets or domains. Also, since the model can handle triple-only scenarios, I suggest the authors also consider reporting the performances on Freebase and Nell as done in query2box, as such, we can see whether or not this model can deal with more basic cases.
>
> We agree with the Reviewer that more datasets will make our proposed approach more convincing. However, to the best of our knowledge at the time of writing this response, Wikidata is the only sufficiently populated and publicly available option for working with hyper-relational KG query answering suitable for our work. One of the issues with other datasets is that the amount of qualifiers relations used is not sufficiently large. We also considered YAGO-4 but the data are based on data from Wikidata with the addition of hierarchical typing information, and, hence, would not be significantly different.
>
> We do expect this to change in the near future. Apart from the fact that more and more public KG's opt for hyper-relational paradigms, it is also becoming more popular in the industry. In parallel, we see the support of hyper-relations by popular triplestores (e.g. GraphDB, Stardog), while it is the organic way of modelling data in property graphstores (e.g. Neo4J, Tigergraph).
>
> We have further observed that hyper-relations are prefered in different domains (e.g modelling biomedical data), as they allow for qualification of interactions within interaction networks e.g. Drug A regulates Protein A, effect: up/down regulation.
>
> As to the triple-only baselines, we confirm that StarQE is backward compatible with triple-only graphs. In fact, in the triple-only setup our method resembles MPQE, but uses a different GNN encoder (CompGCN in our case vs. R-GCN for MPQE). We probe that in the experiments on the triple-only version of the new datasets (Table 1).
> Running with the additional datasets would not really give more insight as to how the qualifiers help, but rather about the basic performance of MPQE.
>
>
> >The novelty is limited. The model is just a combination of two existing approaches.
>
> The focus of our work is different from proposing models. The encoder that can process hyper-relational graphs is not listed as one of our main contributions. Instead, we focus our efforts on formalizing the problem of hyper-relational query embedding, linking the problem to the FOL roots, and proposing a relatively simple method to answer hyper-relational queries by combining appropriate parts of existing approaches. While we do acknowledge the adaptation of the StarE GNN encoder, we would like to emphasize that our proposed method could work with any other encoder supporting hyper-relational KGs, if any were to become available.

---

> > ### Author Response · Authors · 2021-11-17
> > **Response to iDXR Part II**
> >
> >
> > >It cannot handle queries about numbers, text, and time. While some existing methods [a][b][c] can handle such information in knowledge graph embeddings. Those important references are missing and should be used as baselines especially on 1p queries.
> >
> > We thank the Reviewer for the pointers and do agree that including literals is a promising idea (mentioning this in Section 6 of the manuscript). Generally speaking, even 1p link prediction is quite non-trivial in the presence of literals. The approaches [a] and [b] you listed only augment decoder-only models which are not applicable in the query embedding domain. Moreover, they are quite limited in the representation capabilities, i.e., [a] only tackles numbers with a regression on top of TransE loss, and [b] essentially treats any literal as a new embedding vector which loses all the semantic similarity behind "close" literal values. The reference [c] is merely a decoder-only model for hyper-relational graphs which can only work with entities and relations, literals are not supported. Besides, as the decoder-only model, it can not be used in query embedding, and is already outperformed by StarE in link prediction qualities in the respective paper. For query embedding task, one would first need a proper __encoder__ model, and to the best of our knowledge of the literature, there exist no GNN encoder for hyper-relational KGs supporting literals.
> >
> > As to the query embedding task, none of the state of the art query embedding approaches support text, numbers, dates, or literals in any other modality. Following the literature, we focus first on answering logical queries over entities in hyper-relational KGs. Extending this framework to literals is possible but presents a challenge worth of its own research paper.
> >
> >
> >
> >
> > > Important reasoning operators such as negation, disjunctions are not considered, making the approach less attractive. I guess we can draw some insights from betaE for this type of operator.
> >
> > In this work, we focus on answering conjunctive hyper-relational queries in latent space. Disjunction and negation are not currently supported and this fact is acknowledged in Section 6 - Limitations & Future Work. As stated in the related work section, the realisation of additional logical operators presents itself as a challenge worthy of their own research papers. That said, supporting more logical operators is definitely in our agenda for future work. Moreover, hyper-relational format allows for new types of operators that are not possible in triple-only systems, e.g., joins over qualifier entities, and we plan to support that as well.
> >
> >
> > > I suggest the authors add some baselines in Table 2 so that we can know the differences between StarQE and baselines in generalization capability.
> >
> >
> > With respect to differences of baselines over generalization capability, we decided to include the MPQE+reif row in Table 2. The reason behind this, is that we are interested in the effects of reification over generalisation. Furthermore, it is the theoretically hardest setup.
> >
> > Additional comparisons with other baselines introduced in Table 1, would not be valid, since they do not consider qualifier information at all (e.g triple-only).
> >
> >
> > > Can the model benefit from nonlinear layers (transformations)?
> >
> > The current model benefits from nonlinear transformations in the form of non-linear activation functions in between the layers (cf. second-to-last sentence in *The Message-Passing Step*, denoted as $\sigma$).

---

> ### Author Response · Authors · 2021-12-03
> **Further Questions**
>
> Dear Reviewer, please let us know if our response addressed your concerns or if you have any further questions.

---

### Official Review · Reviewer_frRt · 2021-11-06

**Correctness:** 3
**Technical Novelty And Significance:** 3
**Empirical Novelty And Significance:** 3
**Recommendation:** 6
**Confidence:** 4

**Main Review:**

The paper is well written and is an important step in the direction of handling multiple representations of knowledge graphs. Most existing approaches are focused on triple based techniques, however ignoring complex but more expressive representations such as reification and/or qualifiers. Therefore, in terms of novelty of the problem, this work clearly seems to be taking the next step in query embeddings. The approach is an interesting and valid combination of other approaches to solve the problem. The evaluation of the dataset clearly shows that the approach works.

My main concerns of this work is as follows:
1. The aspect of entering a new territory should also be backward compatible with older representations. This raises the question of the performance of the system on Freebase datasets primarily evaluated by other state of the art approaches such as BetaE, Query2Box, EmQL etc.
2. EmQL evaluates faithfulness of the overall approach. It is also important to measure how well the work performs on complete knowledge graphs and not only incomplete ones. Have the authors tried to do this?
3. The change from Query2Box, EMQL to Beta Embeddings was its way of handling negation and disjunction operators. How does this work compare in terms of handling the operators of EFOL?
4. Why does Oracle's performance drop for 3p queries and how does the proposed approach handle them better? If there are qualitative analysis on this it would be great.
5. Can you please elaborate on StarE-like row in Table 2? I am not convinced that it is an apples to apples comparison to the other rows in the table.


**Summary Of The Paper:**

The paper presents an approach for embeddings queries over hyper relation graphs. Hyper-relational graphs, specifically those that have context are usually represented and encoded as regular triples by the existing approaches. To do so, the work represent the hyper-relational graphs as FOL with parameterizing the predicate to include context. The model is a combination of three architectures where firstly encodes the query graph using StarE, then CompGCN to aggregate for representing qualifiers, and MPQE for learning the query embeddings. In order to evaluate this approach, the work defines a new dataset on Wikidata because of reified triples/qualifiers.

**Summary Of The Review:**

The paper is well written and addresses an important direction is handling more complex representations of knowledge graphs. The work seems solid however lacks some aspects of evaluation. It is unable to justify that it is backward compatible to the original datasets which would be of great value even though not being at par with the state of the art approaches. In other words, it is ok to show drop in performance on Freebase datasets but claim generalizability over both traditional representations and KGs with reification.

---

> ### Author Response · Authors · 2021-11-17
> **Response to frRt**
>
> We would like to thank the Reviewer for their feedback. We are pleased to see the acknowledgment of the novelty and importance of our work as the next step for the field. Regarding the Reviewer's concerns:
>
> > **W1**  This raises the question of the performance of the system on Freebase datasets primarily evaluated by other state of the art approaches.
>
> This is an important point and we confirm that StarQE is backward compatible with triple-only graphs. In fact, in the triple-only setup our method resembles MPQE, but uses a different GNN encoder (CompGCN in our case vs. R-GCN for MPQE). We probe that in the experiments on the triple-only version of the new datasets (Table 1).
>
> > **W2.** EmQL evaluates faithfulness of the overall approach.
>
> We agree that faithfulness is an interesting aspect of query embedding models, but for this paper, it fell out of scope due to the lack of space. Nevertheless, we performed faithfulness experiments by evaluating our system on the _training set_ as done in the related EmQL paper.
>
> We report our results over the __Hits@3__ metric, for consistency with the EmQL paper. We included standard deviation across 5 runs with different random seeds.
>
> |             | 1p               | 2p               | 3p               | 2i               | 3i               | 2i-1p            | 1p-2i            |
> |:------------|:-----------------|:-----------------|:-----------------|:-----------------|:-----------------|:-----------------|:-----------------|
> | __StarQE-like__ | $84.06 \pm 4.64$ | $98.12 \pm 0.49$ | $96.56 \pm 0.89$ | $97.48 \pm 1.15$ | $99.81 \pm 0.11$ | $98.78 \pm 0.51$ | $99.84 \pm 0.09$ |
> | __MPQE-like__   | $96.18 \pm 1.10$ | $14.19 \pm 1.18$ | $32.12 \pm 2.59$ | $97.17 \pm 0.62$ | $99.46 \pm 0.16$ | $18.64 \pm 1.43$ | $85.42 \pm 1.24$ |
>
> As we can observe from the results of the faithfulness experiments, our StarQE model is _faithful_ in terms of entailment over the training set. However, we can observe how we lose faithfulness when training only on 1p queries but evaluating on all (MPQE-like), which falls in line with the expected results. It is interesting to observe how impactful different training regimes are on faithfulness.
>
> We also included the full experimental results into the Appendix K of the updated paper, cf Table 5.
>
> > **W3.** The change from Query2Box, EMQL to Beta Embeddings was its way of handling negation and disjunction operators. How does this work compare in terms of handling the operators of EFOL?
>
> In this work, we focus on answering conjunctive hyper-relational queries in latent space. Disjunction and negation are not currently supported and this fact is acknowledged in Section 6 - Limitations & Future Work. As stated in the related work section, the realisation of additional logical operators presents itself as a challenge worthy of their own research papers. That said, supporting more logical operators is definitely in our agenda for future work. Last but not least, hyper-relational format allows for new types of operators that are not possible in triple-only systems, e.g., joins over qualifier entities, and we plan to support that as well.
>
> > **W4.** Why does Oracle's performance drop for 3p queries and how does the proposed approach handle them better?
>
> The Oracle has access to train, validation, and test data but does not have access to the qualifier information (we elaborate more on that setup in Appendix I).
> Our intuition to the observed behavior is that the oracle might make mistakes (where qualifier-aware approach would not) which can accumulate over multiple hops. Interestingly, this observation is consistent with the related work over triple-only graphs such as CQD [1].
>
> [1] Arakelyan et al. Complex query answering with neural link predictors. ICLR 2021.
>
> > **W5.** Can you please elaborate on StarE-like row in Table 2? I am not convinced that it is an apples to apples comparison to the other rows in the table.
>
> The terms StarE-like, Q2B-like, EmQL-like, and MPQE-like refer to the *training regimes* and not to the particular *model*. In fact, all rows use the same model StarQE (except for the MPQE reified one), but different sets of training queries (the ones in yellow). We evaluate the respective trained models on all query shapes to test for generalisation capabilities, i.e., whether it is necessary to have a certain query shape in the training set in order to answer these types of queries. Similar experiments have been conducted in literature which differ in the training query set. We refer to these settings by the name of paper which first used it. "StarE-like" just means that we used all query types in the training set.
>
> We intended to use the term “StarQE-like”, but accidentally left the 'Q' out. we updated this in the paper.
> Perhaps the use of bold font is confusing to readers as it can be understood as implying a comparison between different models. If the Reviewer agrees, we would remove it in the revised version of the manuscript.

---

> ### Author Response · Authors · 2021-12-03
> **Further Questions**
>
> Dear Reviewer, please let us know if our response addressed your concerns or if you have any further questions.

---

### Official Review · Reviewer_wJVm · 2021-11-06

**Correctness:** 3
**Technical Novelty And Significance:** 3
**Empirical Novelty And Significance:** 3
**Recommendation:** 6
**Confidence:** 3

**Main Review:**

Strength:
- The paper is well-motivated, and the authors propose a reasonable solution to the hyper-relational KG QE problem.
- Experiments show that the proposed problem outperforms most of the previous methods that only use triple-only graphs.
- The authors prepare a new dataset for the hyper-relational KG QE problem.


Weakness:
- The description of their proposed method is not very clear. For example, "we use X[y] to indicate the representation of y in X" in the Qualifier Representation paragraph of Sec 4. Readers may be confused about the meaning of “X”. The authors should explain that “X” can be E, R, etc.
- In table 1, compared with Reification, the performance improvement of StarQE is not very obvious.


Questions:
- Could you clarify whether “StarE-like” in table 2 is the same as “StarQE” in table 1? It is better to use a more consistent name or explain their relation more clearly in Table 2 caption.


**Summary Of The Paper:**

In this paper, the authors propose a framework to learn the query embeddings for hyper-relational KGs, which allows QA over hyper-relational KGs with more complex questions and makes use of the qualifiers. They also introduce a new hyper-relational KG QA dataset, WD50K-QE, for the evaluation of their proposed method. Experiments show that qualifiers help their framework achieve more accurate results in general compared with the baselines that use information from triple-only graphs.


**Summary Of The Review:**

---

> ### Author Response · Authors · 2021-11-17
> **Response to wJVm**
>
>
> We would like to thank the Reviewer for their comments. It is great to see the soundness and motivation of our work being recognized alongside the creation of a new dataset to support our experiments.
>
> > **W1.** The description of their proposed method is not very clear. For example, "we use X[y] to indicate the representation of y in X" in the Qualifier Representation paragraph of Sec 4. Readers may be confused about the meaning of “X”. The authors should explain that “X” can be E, R, etc.
>
> We thank the reviewer for the constructive suggestion. We have updated the paragraph as per Reviewer's recommendation with an example to make it more clear which matrix is being indexed at what time.
>
>
>
> > **W2.** In table 1, compared with Reification, the performance improvement of StarQE is not very obvious.
>
> The fact that reification does show similar performance to the qualifier embedding case, shows that our model is __robust__ to the representation of hyper-relational facts and does not lose the _semantic_ information encoded in reified or hyper-relational facts. That is, we would rather treat this fact as a positive signal.
> Note that reification is not a baseline which we aim to beat but rather, a different way of representating hyper-relational information that comes with specific costs and benefits (e.g. the change in query structure, different structural properties of the query graph, etc.).
>
>
> > **Question.** Could you clarify whether “StarE-like” in table 2 is the same as “StarQE” in table 1? It is better to use a more consistent name or explain their relation more clearly in Table 2 caption.
>
>
> The terms StarE-like, Q2B-like, EmQL-like, MPQE-like, and MPQE-like + Reif refer to the *training regimes* and not to the particular *model*. In fact, all rows use the same model, but different sets of training queries (the ones in yellow). We evaluate the respective trained models on all query shapes to test for generalisation capabilities, i.e., whether it is necessary to have a certain query shape in the training set in order to answer these types of queries. Similar experiments have been conducted in literature which differ in the training query set. We refer to these settings by the name of paper which first used it. "StarE-like" just means that we used all query types in the training set.
>
> We intended to use the term “StarQE-like”, but accidentally left the 'Q' out. we updated this in the paper.
>
>
> Perhaps the use of bold font is confusing to readers as it can be understood as implying a comparison between different models. If the Reviewer agrees, we would remove it in the revised version of the manuscript.

---

> > ### Comment · Reviewer_wJVm · 2021-12-01
> > **Thanks for the response**
> >
> > I have read the response and will keep my score the same.

---

### Author Response · Authors · 2021-11-17
**General Response**

We would like to thank the Reviewers for their valuable and insightful comments. We are happy to see the recognition of our work being "well-motivated" (**wJVm**), "seems to be taking the next step in query embedding" (**frRt**), interesting and delivered in a clear, easy-to-read manner (**wJVm**, **frRt**, **iDXR**). Through our experimental design and results, we aspire to highlight the importance and value of bringing qualifier information to query answering.

Furthermore, we are content to see the acknowledgement of our work as an interesting future direction for the field of Query Embedding, as it acted as one of the key motivators behind it.

In addition to that, the creation and introduction of a new (hyper-relational) query dataset was a challenging yet necessary step towards performing our experiments. We are confident that this (public) dataset will be a substantial resource for our field.


For other points raised by the Reviewers and for their convenience, we will address them under the original reviews.
Based on the feedback, we also updated the paper and included a new Appendix K where we investigate how *faithful* the model is.

---

### Author Response · Authors · 2021-11-25
**Discussion**

Dear Reviewers,

thank you for the feedback - it helped us to further improve the manuscript and report a new faithfulness experiment in the revised version.
Please let us know if our clarifications and revisions addressed your concerns.

---

### Comment · Reviewer_GQAR · 2021-11-28
**Comments to the Disucssions and Feedback**

For W1. The challenge of multi-hop logical reasoning must be clearly described, in the form of a reference, which is not persuasive.
For W2. An accurate summary of the experimental results is needed in the abstract.

---

> ### Author Response · Authors · 2021-11-29
> **Response to GQAR**
>
> Given that the reviewer response arrived after the deadline for updating the manuscript, we will include the suggested updates in the camera-ready version.

---

### Decision · Program_Chairs · 2022-01-20

**Decision:**

Accept (Poster)

**Comment:**

This paper presents a query embedding approach for answering multi-hop queries over a hyper-relational knowledge graph (KG). The main contributions are a new dataset (WD50K-QE) for this task and a simple but sensible extension to an existing model for query embeddings to also handle relation qualifiers. Reviewers wJVm and Bute note that the reification and StarQE models perform similarly. While this is not a negative result, as the authors note, it does raise the question of the relative pros and cons of the two methods. I hope the authors can add a discussion of when one might prefer StarQE over the conceptually simpler reification method in the final version. The authors addressed Reviewer frRt’s concerns about faithfulness and backwards compatibility (though more evidence on purely triple-based tasks would be nice here). Reviewer GQAR also raised some concerns about writing, but the other reviewers mostly found the paper to be well written and well motivated and I tend to agree. Overall, while there are some very good suggestions on how the paper can be extended and improved, I find the current contributions to be substantial enough to warrant a publication.